# End-to-End Learning Framework for Solving Non-Markovian Optimal Control

**Xiaole Zhang** [* 1]  **Peiyu Zhang** [* 1]  **Xiongye Xiao** [* 1 2]  **Shixuan Li** [1]
**Vasileios Tzoumas** [3]  **Vijay Gupta** [4]  **Paul Bogdan** [1]

## Abstract

Integer-order calculus fails to capture the long-range dependence (LRD) and memory effects found in many complex systems. Fractional calculus addresses these gaps through fractional-order integrals and derivatives, but fractional-order dynamical systems pose substantial challenges in system identification and optimal control tasks. In this paper, we theoretically derive the optimal control via *linear quadratic regulator* (LQR) for *fractional-order linear time-invariant* (FOLTI) systems and develop an end-to-end deep learning framework based on this theoretical foundation. Our approach establishes a rigorous mathematical model, derives analytical solutions, and incorporates deep learning to achieve data-driven optimal control of FOLTI systems. Our key contributions include: (i) proposing a novel method for system identification and optimal control strategy in FOLTI systems, (ii) developing the first end-to-end data-driven learning framework, **F**ractional-**O**rder **L**earning for **O**ptimal **C**ontrol (FOLOC), that learns control policies from observed trajectories, and (iii) deriving theoretical bounds on the sample complexity for learning accurate control policies under fractional-order dynamics. Experimental results indicate that our method accurately approximates fractional-order system behaviors without relying on Gaussian noise assumptions, pointing to promising avenues for advanced optimal control.

*Equal contribution [1]Ming Hsieh Department of Electrical and Computer Engineering, University of Southern California, Los Angeles, CA 90089, USA [2]Min H. Kao Department of Electrical Engineering and Computer Science, University of Tennessee, Knoxville, TN 37996, USA [3]Department of Aerospace Engineering, University of Michigan, Ann Arbor, MI 48109, USA [4]Elmore Family School of Electrical and Computer Engineering, Purdue University, West Lafayette, IN 47907, USA. Correspondence to: Xiaole Zhang <xiaolezh@usc.edu>.

*Proceedings of the $42^{nd}$ International Conference on Machine Learning*, Vancouver, Canada. PMLR 267, 2025. Copyright 2025 by the author(s).

## 1. Introduction

Many real-world systems, including biological systems, neuronal networks, organ networks, financial markets, and cyber-physical systems, exhibit long-range dependence (LRD) and memory effects that traditional integer-order models struggle to capture (Lundstrom et al., 2008; Ivanov et al., 1999; Ghorbani & Bogdan, 2013; Ghorbani et al., 2018). Fractional-order linear time-invariant (FOLTI) systems extend classical integer-order models through fractional calculus, providing a more flexible framework for modeling such complex behaviors. Due to their ability to represent non-local dependencies and history-dependent dynamics, FOLTI systems have found broad applications in control systems (Baleanu et al., 2011; Monje et al., 2010; Reed et al., 2023b), bioengineering (Magin, 2004; 2010; Ghorbani & Bogdan, 2013), neuroscience (Pequito et al., 2015; Gupta et al., 2019; Reed et al., 2022; 2023a), cyber-physical systems (Xue & Bogdan, 2017), and financial modeling (Granger & Joyeux, 1980). However, the practical deployment of FOLTI systems faces significant challenges in system identification and optimal control. Unlike Markovian systems, where the current state depends solely on the immediate past state, FOLTI systems exhibit non-Markovian properties, meaning their evolution is influenced by a wide range of past states. This memory effect introduces additional complexities, making it difficult for traditional methods to generalize effectively to FOLTI systems.

Despite their advantages, FOLTI systems present significant challenges in both modeling and control. Identifying system parameters for fractional-order dynamical systems remains an open research problem. One major difficulty in system identification arises from the non-Markovian nature of fractional-order systems, where the state evolution is not solely determined by recent states but depends on an entire history of past observations. Additionally, the fractional derivative operators, such as the Grünwald–Letnikov fractional derivative (Hilfer, 2000; Monje et al., 2010; Oldham & Spanier, 1974), introduce combinatorial complexity and nonlinearities, making standard parameter estimation techniques ineffective. Existing methods often rely on strong assumptions, such as noiseless data or constrained data gen-

eration processes, limiting their applicability in practical scenarios.

Beyond system identification, optimal control of FOLTI systems remains a major challenge. Unlike integer-order systems, where well-established control frameworks exist, fractional-order systems exhibit long-range dependencies and memory effects, making classical control strategies inadequate. In particular, the lack of analytical solutions for optimal control in FOLTI systems requires new approaches that account for their unique dynamics. Addressing these challenges is crucial for enabling real-world applications of FOLTI systems in control systems.

A well-established framework for optimal control in integer-order systems is the Linear Quadratic Regulator (LQR), introduced by Rudolf Kalman in 1960 (Kalman et al., 1960). LQR determines an optimal control policy that minimizes a quadratic cost function defined over the system state and control input, subject to weighing matrices, for systems governed by linear dynamics. The traditional integer-order LQR problem has been extensively studied over the decades (Dean et al., 2018; 2020; Tu & Recht, 2019). For linear time-invariant (LTI) systems with known parameters, the LQR problem has a closed-form solution on the infinite time horizon and can be solved efficiently using dynamic programming on finite horizons (Anderson & Moore, 2007). Furthermore, numerous researchers have developed end-to-end guarantees for LQR in the context of LTI systems. These guarantees often involve a two-step process: (i) estimating the unknown system parameters, and (ii) designing robust controllers to account for model uncertainties.

While the LQR problem for integer-order LTI systems is well understood (see related work in Appendix B), far less attention has been devoted to the case of fractional-order dynamical systems (Monje et al., 2010). These systems, which originate from fractional calculus (Hilfer, 2000; Ionescu et al., 2017), differ significantly from their integer-order counterparts. Unlike Markovian systems, where the current state depends solely on the immediate past state, fractional-order systems exhibit a non-Markovian behavior, with the current state influenced by a memory effect spanning a broad range of past states. This memory effect makes the analysis and control of fractional-order dynamical systems particularly challenging. Estimation of unknown parameters in fractional-order dynamical systems remains an open problem (Yaghooti & Sinopoli, 2023; Chatterjee & Pequito, 2022; Zhang et al., 2025), even under the assumption of linear fractional operators. For finite horizons, optimal control problems have been addressed under specific assumptions (Li & Chen, 2008), but no comprehensive results exist for infinite horizons. Moreover, achieving end-to-end guarantees for fractional-order systems is particularly challenging, as both system identification (Yaghooti & Sinopoli, 2023;

Zhang et al., 2025) and robust control present significant technical difficulties.

To address these challenges, this work establishes a theoretical framework for fractional-order LQR, deriving a principled end-to-end learning approach for FOLTI optimal control. Specifically, we formulate the mathematical structure of FOLTI systems, enabling a rigorous extension of LQR to the fractional-order setting. Given the diagonal elements of the system matrix, we propose a method to estimate the unknown system parameters from multiple observed trajectories. Using this estimated model, we derive the optimal control policy using LQR, leveraging least squares optimization and Lagrange multipliers to obtain analytically tractable solutions despite the inherent complexities introduced by fractional-order dynamics. We further analyze the sample complexity of this learning process, quantifying the number of samples required for the estimated LQR loss to converge to the true LQR loss under unknown input conditions.

Beyond theoretical derivations, we develop an end-to-end data-driven learning framework, fractional-order learning for optimal control (FOLOC), that jointly performs system identification and optimal control. In contrast to classical control methods that require pre-specified system dynamics, our approach learns both the system parameters and control policies directly from observed trajectories. Unlike classical approaches that assume structured noise distributions, our framework is designed to operate in realistic, noisy environments. Specifically, we incorporate deep learning-based modeling techniques to ensure that the control policy remains robust under non-Gaussian noise, distributional shifts, and real-world uncertainties.

In summary, this work makes the following key contributions: (i) We provide theoretical foundations for optimal control of fractional-order linear time-invariant (FOLTI) systems. (ii) We develop a principled end-to-end data-driven learning framework, FOLOC, that jointly optimizes system identification loss and control policies, ensuring robustness to non-Gaussian noise and real-world uncertainties. (iii) We analyze the theoretical sample complexity, providing convergence guarantees for learned control policies under unknown system dynamics. (iv) We conduct extensive experiments, demonstrating the FOLOC's robustness across noise distributions, scalability to different system complexities, and computational efficiency in real-time control tasks.

## 2. Preliminaries and Problem Formulation

In this section, we introduce key theoretical foundations from fractional calculus, focusing on the Grünwald–Letnikov fractional-order derivative and its role in modeling FOLTI systems. By extending LQR to FOLTI systems, we then formulate the end-to-end learning problem,

where the objective is to identify system parameters and optimize control policies directly from data. This formulation integrates system identification and control optimization, addressing challenges arising from memory effects and non-Markovian dynamics in FOLTI systems.

## 2.1. Fractional-Order Derivative

**Definition 2.1** (Grünwald–Letnikov fractional-order derivative)**.** The left-side and right-side Grünwald–Letnikov fractional-order derivatives of order $\alpha$ on the finite interval $I = [a, b]$ are defined as follows:

$$\left({}^{GL}D_{a+}^{\alpha}f\right)(x) = \lim_{h \to 0^+} \frac{1}{h^{\alpha}} \sum_{k=0}^{\lfloor \frac{x-a}{h} \rfloor} (-1)^k \binom{\alpha}{k} f(x - kh),$$
(1)

$$\left({}^{GL}D_{b-}^{\alpha}f\right)(x) = \lim_{h \to 0^-} \frac{1}{h^{\alpha}} \sum_{k=0}^{\lfloor \frac{b-x}{h} \rfloor} (-1)^k \binom{\alpha}{k} f(x + kh),$$
(2)

where $x \in \mathbb{R}$, $0 < h < b - a$, and $\alpha > 0$. Here, $\binom{\alpha}{k}$ denotes the generalized binomial coefficient, defined as $\binom{\alpha}{k} = \frac{\Gamma(\alpha+1)}{\Gamma(k+1)\Gamma(\alpha-k+1)}$, where $\Gamma(\cdot)$ denotes the gamma function, which is defined as $\Gamma(z) = \int_0^{\infty} t^{z-1} e^{-t} \, dt$ for $z > 0$.

## 2.2. Grünwald–Letnikov Difference Operator

The fractional-order Grünwald–Letnikov difference operator allows to discretize the fractional-order derivative and represent it as a finite difference as follows:

$$\Delta^{\alpha} x_k := \sum_{j=0}^{k} D(\alpha, j) x_{k-j},$$
(3)

where $x_k \in \mathbb{R}^n$, $\alpha = [\alpha_1, \alpha_2, \ldots, \alpha_n]^\top \in \mathbb{R}^n$ represents the order of the difference operator, and $D(\alpha, j) \in \mathbb{R}^{n \times n}$ is an $n \times n$ diagonal matrix defined as

$$D(\alpha, j) := \text{diag}\left(\psi(\alpha_1, j), \psi(\alpha_2, j), \ldots, \psi(\alpha_n, j)\right),$$
(4)

with

$$\psi(\alpha_i, j) := \frac{\Gamma(j - \alpha_i)}{\Gamma(-\alpha_i)\Gamma(j+1)}, \quad i = 1, 2, \ldots, n.$$
(5)

## 2.3. Fractional-Order Linear Time Invariant System

The state-space representation of the discrete-time fractional-order linear time invariant system reads:

$$\Delta^{\alpha} x_{k+1} = A x_k + B u_k,$$
(6)

where $x_k \in \mathbb{R}^n$ is the state vector, $u_k \in \mathbb{R}^m$ is the system input, matrices $A$ and $B$ are constant matrices with size $n \times n$ and $n \times m$, respectively.

Using the Grünwald–Letnikov difference operator, we can write the system as follows:

$$x_{k+1} = A x_k + B u_k - \sum_{j=1}^{k+1} D(\alpha, j) x_{k+1-j}.$$
(7)

The last term in Eq. 7 represents the memory-dependent nature of fractional-order dynamical systems, making them suitable for modeling non-Markovian and long-range dependent processes.

## 2.4. Linear Quadratic Regulator for FOLTI systems

The linear quadratic regulator (LQR) problem concerns the optimal control of a dynamical system given known and fixed quadratic costs. Formally, the goal of LQR is to find the optimal control $u_t$ that minimizes the quadratic loss fitting in the FOLTI systems, which in turn solves the following optimization problem.

**Definition 2.2** (LQR for FOLTI systems)**.** The LQR problem for FOLTI systems is defined as follows:

$$\min_{\{u_i\}_{i=0}^{T-1}} J_T(u) := \sum_{k=0}^{T-1} \left(x_k^\top Q x_k + u_k^\top R u_k\right) + x_T^\top Q_f x_T$$
(8)

$$\text{s. t. } \Delta^{\alpha} x_{k+1} = A x_k + B u_k,$$
(9)

where $Q \in \mathbb{R}^{n \times n}$ and $Q_f \in \mathbb{R}^{n \times n}$ are positive semi-definite matrices representing the state cost and terminal state cost, respectively, where $n$ is the dimension of the state vector $x_k$. Similarly, $R \in \mathbb{R}^{m \times m}$ is a positive semi-definite matrix representing the control cost, where $m$ is the dimension of the control input vector $u_k$.

## 2.5. Problem Formulation

Consider a fractional-order linear time-invariant system defined using the Grünwald–Letnikov difference operator. The goal is to find an optimal control sequence via solving LQR problem based on system identification from observed data.

Specifically, we are given $N$ trajectories, each trajectory consisting of $T$ time steps, sampled from a joint distribution $\Pi$ over initial states, process noise, system parameters and cost matrices, i.e., $\{x_0, \{w_k\}_{k=0}^{T-1}, \{u_k\}_{k=0}^{T-1}, \{A, B, \alpha\}, \{Q, R\}\} \sim \Pi$. Thus, we can represent the dataset as $\mathcal{D} = \{\{x_k^i\}_{k=0}^{T}, \{u_k^i\}_{k=0}^{T-1}, Q^i, R^i, \{u_k^{i\,o}(x_0^i, A, B, \alpha)\}_{k=0}^{T-1}\}_{i=1}^{N}$, where $u_k^{i\,o}$ denotes the optimal control input. We aim to solve the following two-step problem:

**(i) System Identification.** The fractional-order system dynamics are parameterized by $\Theta := \{A, B, \alpha\}$, where $A \in \mathbb{R}^{n \times n}$ and $B \in \mathbb{R}^{n \times m}$ are system matrices, and

$\alpha \in (0,1)^n$ is the fractional order. The system identification problem is posed as:

$$\hat{\Theta} = \arg \min_{\Theta} \sum_{i=1}^{N} \sum_{k=1}^{T} \left\| \hat{x}_k^i(\Theta) - x_k^i \right\|_2^2, \qquad (10)$$

where $\hat{x}_k^i(\Theta)$ represents the predicted state at time $k$ for the $i$-th trajectory, obtained using the identified parameters $\Theta$.

**(ii) Optimal Control.** Once $\hat{\Theta} = \{\hat{A}, \hat{B}, \hat{\alpha}\}$ is estimated, the optimal control problem is formulated as:

$$\min_{\{u_k\}_{k=0}^{T-1}} \hat{J}_T(u) := \sum_{k=0}^{T-1} \left( x_k^\top Q x_k + u_k^\top R u_k \right) + x_T^\top Q_f x_T,$$
$$(11)$$

subject to the identified system dynamics:

$$\Delta^{\hat{\alpha}} x_{k+1} = \hat{A} x_k + \hat{B} u_k. \qquad (12)$$

**Deep Learning-Based Reformulation.** The optimal control problem can be framed in a deep learning context as learning an operator $\mathcal{L} : \mathcal{D} \to \mathcal{U}$, where $\mathcal{U}$ represents the set of optimal control inputs. Formally, the operator $\mathcal{L} : \mathcal{D} \to \mathcal{U}$ computes the optimal control policy $\{u_t^o\}_{t=0}^{T-1}$ directly from the observed trajectory, while simultaneously uncovering the system parameters $\Theta = \{A, B, \alpha\}$ that encapsulate the dynamics of the fractional-order system.

## 3. Methodology

### 3.1. Mathematical Foundations for Fractional-Order Learning and Control

**Lemma 3.1** (Discrete-time FOLTI system solution). *The solution to the discrete-time FOLTI system is given by (Guermah et al., 2012):*

$$x_k = G_k x_0 + \sum_{j=0}^{k-1} G_{k-1-j} B u_j, \qquad (13)$$

*where the matrices $G_k$ are defined recursively as:*

$$G_k = \begin{cases} I & \text{for } k = 0, \\ \sum_{j=0}^{k-1} A_j G_{k-1-j} & \text{for } k \geq 1, \end{cases} \qquad (14)$$

*and the matrices $A_j$ are given by:*

$$A_j = \begin{cases} A - diag(\alpha_1, \ldots, \alpha_n) & \text{if } j = 0, \\ -D(\alpha, j+1) & \text{if } j \geq 1. \end{cases} \qquad (15)$$

The proof is provided in Appendix D.2. The first component of the FOLTI system solution represents the unforced response of the system. The term $G_k$ exhibts the particularity of being time-varying, attributed to the fractional-order $\alpha$ which inherently accounts for all the past states. The second component takes the role of the convolution sum corresponding to the forced response.

**Theorem 3.2** (LQR solution for FOLTI systems). *The least-squares optimal control solution for the LQR problem of a FOLTI system is given by:*

$$U = -\left( G^\top \bar{Q} G + \bar{R} \right)^{-1} G^\top \bar{Q}^\top H x_0, \qquad (16)$$

*whereas the Lagrange multiplier optimal control solution is:*

$$U = -R^{-1} B^\top \otimes (I - G_\lambda)^{-1} H_\lambda x_0. \qquad (17)$$

*Here, $U$, $G$, $G_\lambda$, $H$, $H_\lambda$, $\bar{R}$, and $\bar{Q}$ are defined in Appendix D.3, and $\otimes$ denotes the Kronecker product. Notably, $U = \begin{bmatrix} u_0^\top & u_1^\top & \cdots & u_{T-1}^\top \end{bmatrix}^\top$.*

Using the optimal control solution via LQR, a natural question arises: given a system identification algorithm and a threshold error $\epsilon$, how many samples are required for the system identification process to ensure that the error between the estimated LQR loss $\hat{J}$ and the true LQR loss $J$ (computed with the true system parameters) remains within the threshold $d(\hat{J} - J) \leq \epsilon$, where $d(\cdot)$ denotes the metric. We derive the following sample complexity results.

**Theorem 3.3** (Sample Complexity for FOLTI systems). *Consider a FOLTI system with a known matrix $A$, fractional order $\alpha$, and an unknown matrix $B$. The following sample complexity bounds hold using the system identification in Appendix D.1: Let*

$$K_B = \frac{1}{N} (\phi^\top \phi)^{-1} \phi^\top K_w \phi (\phi^\top \phi)^{-1},$$

*(a) Least-squares solution.*

$$\mathbb{E}\left[ \left| \hat{J} - J \right| \right] \leq \|z\|_2^2 \|R^{-1}\|_2 \left( 1 + \|B\|_2^2 \|R^{-1}\|_2 \|S\|_2 \right)$$
$$\cdot \left( \text{Tr}(K_B) + 2\|B\|_2 \sqrt{\text{Tr}(K_B)} \right), \quad (18)$$

*(b) Lagrange multiplier solution. Assume $L_{QG}^{-1}(I - L_u) \succeq 0$,*

$$\mathbb{E}\left[ \left| \hat{J} - J \right| \right] \leq \|z\|_2 \|H_\lambda x_0\|_2 \|R^{-1}\|_2 \|\mathbb{L}\|_2 \left( 1 + \|B\|_2^2 \right.$$
$$\left. \cdot \|R^{-1}\|_2 \|\mathbb{L}\|_2 \|L_{QG}\|_2 \right) \left( \text{Tr}(K_B) + \right.$$
$$\left. 2\|B\|_2 \sqrt{\text{Tr}(K_B)} \right), \qquad (19)$$

*where $z = G_d^\top \bar{Q}^\top H x_0$, $S = G_d^\top \bar{Q} G_d$, $\|\mathbb{L}\|_2 = \|(L_{QG})^{-1}\|_2 \|L_{QG}\|_2$, and $G_d$, $\bar{Q}$, and $H$ are system matrices as defined in Appendix D.4.*

**Corollary 3.4** (Simplified Sample Complexity for FOLTI systems). *Let $K_w = \sigma_w^2 I$ and $u_0^i \sim \mathcal{N}(0, I)$. For $p > m + 1$, then:*

$$\text{Tr}(K_B) = \frac{nm\sigma_w^2}{N(p - m - 1)}, \qquad (20)$$

*where $p$ denotes the number of initial conditions, and $N$ denotes the number of trials adding Gaussian noise to fractional-order dynamics from these $p$ initial conditions, and $m$ is the dimension of control inputs (see Appendix D.5).*

The sample complexity results demonstrate that the overall convergence rates of the least-squares solution and the Lagrange multiplier solution are both $\mathcal{O}(\frac{1}{\sqrt{N}})$, differing only in their constant factors.

## 3.2. Theoretical Approach for Fractional-Order System Identification and Optimal Control

Our end-to-end theoretical learning method follows a similar strategy to those used in Markovian processes, specifically for LTI systems. The learning procedure can be summarized in two main steps. Given $N$ noisy trajectories, each of $T$ time steps, and assuming the diagonal elements of the system matrix $A$ are known, we first formulate a least-squares problem to identify the system parameters by using the linearity of the Grünwald–Letnikov difference operator. The state evolution with noise can be represented as

$$\Delta^\alpha x_{k+1} = Ax_k + Bu_k + w_k, \qquad (21)$$

where $\{w_k\}_{k=0}^{T-1}$ is an independent and identically distributed (i.i.d) Gaussian noise process. Using the linearity of the fractional difference operator, the state evolution for each trajectory can be expressed as:

$$x_1 = Ax_0 + Bu_0 + C_\alpha x_0 + w_0 \qquad (22)$$

Using this formulation, we define the following optimization problem across $N$ trajectories

$$\hat{\theta} = \arg\min_\theta \|X - \xi\theta\|_2^2, \qquad (23)$$

with the closed-form solution

$$\hat{\theta} = \left(\xi^\top \xi\right)^{-1} \xi^\top X, \qquad (24)$$

where system parameters $C_\alpha, \theta$ and feature matrices $X, \xi$ are detailed in Appendix D.1.

In the second step, the derived system parameters are used to compute the optimal control solution as described in Theorem 3.2. For a given identified FOLTI system and a time horizon $T$, the optimal control sequence can be derived for any cost matrices $Q$ and $R$, along with the initial condition $x_0$.

**Remark.** Knowing the diagonal elements of the system matrix $A$ is a strong assumption and represents a limitation of the theoretical system identification approach. It serves a similar role to assumptions commonly made in classical system identification, where the exact (integer) order of the system is often presumed known in advance. In the context of fractional-order dynamical systems, estimating the system order—i.e., the fractional derivatives—is substantially more challenging, and to the best of our knowledge, no existing method provides theoretical guarantees in a general setting. To address this limitation, we propose FOLOC framework, which can give accurate optimal control policies without this assumption.

### 3.3. Fractional-Order Learning for Optimal Control Framework

Our end-to-end deep learning framework FOLOC is guided by the mathematical derivation of the optimal control policy for FOLTI systems (see Methodology 3.1). Generally, FOLOC framework can be structured into two main components.

**System Identification Module:** Inspired by traditional LTI and our proposed FOLTI system identification, we use deep learning based time-series models to estimate system parameters and approximate intermediate variables. The objective is to learn the underlying function that maps trajectories to the variables required for solving the LQR problem.

Theoretical system identification for both LTI and FOLTI systems predominantly relies on least-squares methods, which can be viewed as operators learning a function that maps from the trajectory and control input space $\mathcal{X}$ to the system parameter space $\mathcal{S}$. Consider the temporal nature of the trajectory, we choose a recurrent neural network to estimate the system parameters. The mapping is defined as $\mathcal{R}_\theta : \mathcal{X} \in \mathbb{R}^{T \times (n+m)} \rightarrow \mathcal{S} \in \mathbb{R}^{n^2+nm+n}$ such that $vectorize(\hat{A}_\theta, \hat{B}_\theta, \hat{\alpha}_\theta) = \mathcal{R}_\theta(\{x_i, u_i\}_{i=0}^{T-1})$. From Lemma 3.1, the function mapping between the system parameters to the time-varying matrices $A_k$ and then to the matrices $G_k$ can be represented as follows respectively

$$\begin{aligned} A_k &= g_k(A, \alpha; \Gamma), \\ G_k &= f_k(A_0, \cdots, A_{k-1}), \end{aligned} \qquad (25)$$

where $g_k(\cdot; \Gamma)$ is obtained as a function of the Gamma function, which governs the fractional-order dynamics. To model the first mapping, we use MLPs with injecting time embeddings to approximate the function $\mathcal{R}_g : \mathcal{S} \rightarrow \mathcal{H} \in \mathbb{R}^{T \times h}$ such that $\{\hat{A}_{i_g}\}_{i=0}^{T-1} = \mathcal{R}_g(\hat{A}_\theta, \hat{\alpha}_\theta, \{e_i\}_{i=0}^{T-1})$, where $h$ can be chosen based on the dynamics complexity and $\{e_i\}_{i=0}^{T-1}$ represents an embedding of a specific timestamp $i$. The second mapping is approximated by an Encoder-only Transformer $\mathcal{R}_f : \mathcal{H} \rightarrow \mathcal{G} \in \mathbb{R}^{T \times n}$ such that $\{\hat{G}_{i_f}\}_{i=0}^{T-1} = \mathcal{R}_f(\{\hat{A}_{i_g}\}_{i=0}^{T-1})$, where $\hat{G}_{i_f}$ represents the learned approximation of the intermediate parameters $G_k$.

**Optimal Control Module:** Building upon the mathematical derivation of the LQR problem using Lagrange multipliers, we use a neural operator to learn the mapping between the

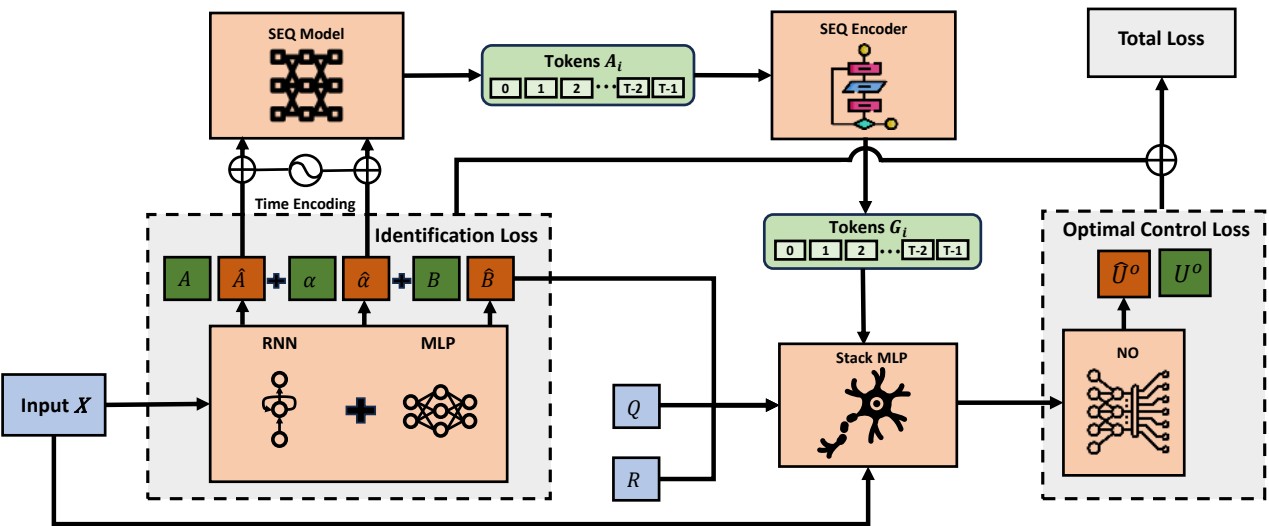

*Figure 1.* Overview of FOLOC framework architecture. The pipeline first infers fractional-order system parameters $(A, \alpha, B)$ from input $\mathcal{X}$ using an RNN+MLP based system identification module. These parameters are then encoded as embeddings of sequential tokens $A_i$ for time-dependent modeling. An attention-based Sequence Encoder processes these embeddings to obtain latent representations, which along with cost matrices $Q, R$, estimated system matrix $\hat{B}$ are fed to the Stack MLPs (see Appendix E.1.4) with residual connection input $\mathcal{X}$ for Fourier Neural Operator to predict optimal control signals. Finally, a composite loss function unifies system identification and control prediction, enabling end-to-end training of both system parameter estimation and control law synthesis.

input space (variable required for solving the LQR problem) and the output space $\{u_i^o\}_{i=0}^{T-1}$ (optimal control sequence).

The core of optimal control module mirrors the process of solving the linear system derived from the LQR problem through Lagrange multipliers. The mathematical formulation of the linear system is expressed as:

$$(I - G_\lambda)\lambda = 2H_\lambda x_0, \quad (26)$$

where $\lambda$ is defined in Appendix D.3 and $I - G_\lambda \in \mathbb{R}^{Tn \times Tn}$ is a block Toeplitz matrix (see Definition D.2). Using the solution of the Lagrange multipliers, the optimal control sequence $\{u_i^o\}_{i=0}^{T-1}$ can be given by:

$$U = -R^{-1}B^\top \otimes (I - G_\lambda)^{-1}H_\lambda x_0. \quad (27)$$

Luo et al. (2024) has demonstrated that solving linear systems via Krylov iterations can be significantly accelerated by using the Fourier Neural Operator (FNO) (Li et al., 2021). Specifically, FNO learns the mapping from the linear system to its corresponding invariant subspace, using the predicted subspace to guide iterative solvers. Inspired by this approach, we use a neural operator to directly learn the mapping between two Hilbert spaces, denoted as $\mathcal{T} : \mathcal{A} \to \mathcal{U} \in \mathbb{R}^{T \times n}$. Due to the block Toeplitz structure of the linear system, the input to FNO is the reduced relevant function variables $a \in \mathbb{R}^{T \times d_a}$. The transformation operator $\mathcal{L}$ first lifts the input $a$ to a higher dimensional channel space $v_0 \in \mathbb{R}^{d_v \times c}$, where $c$ is the number of channels. Then

iteratively applying several Fourier layer to update the representation $v_i \to v_{i+1}$, we have the final $v_T \in \mathbb{R}^{d_v \times c}$, which has the same dimensionality as $v_0$. The FNO outputs optimal control sequence $U = \mathcal{Q}(v_T)$ through the projection of $v_T$ by the transformation operator $\mathcal{Q} : \mathbb{R}^{d_v \times c} \to \mathbb{R}^{T \times n}$. This design enables efficient computation of the optimal control sequence by leveraging the special properties of the block Toeplitz linear system and the expressive capabilities of the FNO.

The loss function used in the training process is defined as follows:

$$\mathcal{L}_t(\theta) = \lambda_w \mathcal{L}_s(\theta) + (1 - \lambda_w) \mathcal{L}_o(\theta), \quad (28)$$

where $\mathcal{L}_s$ quantifies the loss due to system identification errors, and $\mathcal{L}_o$ quantifies the loss due to optimal control errors. Specifically,

$$\mathcal{L}_s(\theta) = \|A - \hat{A}_\theta\|_2^2 + \|B - \hat{B}_\theta\|_2^2 + \|\alpha - \hat{\alpha}_\theta\|_2^2,$$
$$\mathcal{L}_o(\theta) = \|U^o - \hat{U}_\theta^o\|_2^2. \quad (29)$$

Note that $\lambda_w \in [0, 1]$ is a user-defined weighting parameter that determines the balance between system identification and optimal control errors during the learning process.

**Joint optimization pipeline:** The system identification loss $\mathcal{L}_s$ is a physical constraint for FOLOC framework that encourages the model to accurately identify system dynamics, further ensuring accurate control—the ultimate goal of our method. During training, we use system parameters as guidance to ensure accurate system identification, and we do not

need these parameters in testing. Some methods, such as PINN (Raissi et al., 2019), couple multiple loss terms by incorporating physical constraint losses. This helps the system learn representations that are both physically grounded and task-relevant, reducing error propagation and avoiding reliance on purely minimizing prediction error. For real-world datasets where the system parameters are unknown, we can first pretrain the model on synthetic datasets with known parameters $(A, B, \alpha)$. Then, we do not need system parameters (set $\lambda_w = 0$) and finetune the model on the real-world dataset. We provide results for $\lambda_w = 0$ for comparison in Appendix H.2.

**Remark.** The fractional-order dynamical system parameters $A, B$ and $\alpha$ can be extracted from an intermediate layer in the network because of the underlying loss function. Though our end-to-end deep learning framework is guided by the theoretical optimal control solution derived by using Lagrange multipliers, certain learned parameters do not directly correspond one-to-one with their theoretical counterparts (see Appendix E.1 for details).

## 4. Experiments

In this section, we empirically evaluate the proposed system identification algorithm and FOLOC framework using both synthetic and real-world system dynamics. The synthetic data is categorized into two primary types: (i) data generated from a single FOLTI system with fixed system parameters $A, B$, and $\alpha$; and (ii) data generated from multiple FOLTI systems, where $A, B$ are fixed, and $\alpha$ vary across systems. The real-world data is simulated from cart-pole and quadrotor dynamical systems.

### 4.1. Performance Evaluation on Synthetic Data

We evaluate our theoretical end-to-end learning approach and FOLOC framework using synthetic data. Due to the strict assumptions required for the theoretical learning process, we test it using synthetic data generated from a single underlying FOLTI system, with data corrupted by Gaussian noise. Our empirical analysis examines the performance of FOLOC framework from different perspectives.

The underlying fractional-order dynamics used to generate the data for our model are defined as follows:

$$x_{k+1} = Ax_k + Bu_k - \sum_{j=1}^{k+1} D(\alpha, j)x_{k+1-j} + w_k, \quad (30)$$

where $w_k$ represents the process noise.

***Theoretical approach validation.*** By varying the number of trajectories used in the system identification process, we empirically validate Lagrange multiplier based sample complexity results presented in Theorem 3.3 as shown in Fig. 2.

We also compare our method with the traditional end-to-end learning framework for LTI systems (see Appendix I). The mean MSE of FOLTI system learning framework decreases by 81.36% compared to LTI system learning framework.

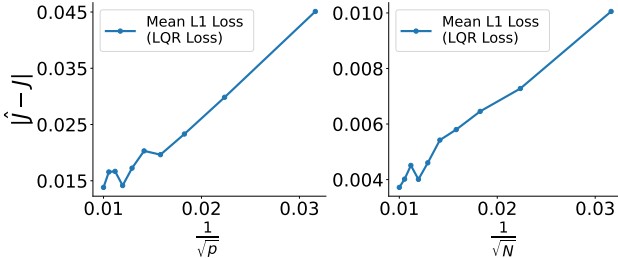

*Figure 2.* Lagrange multiplier based sample complexity simulation.

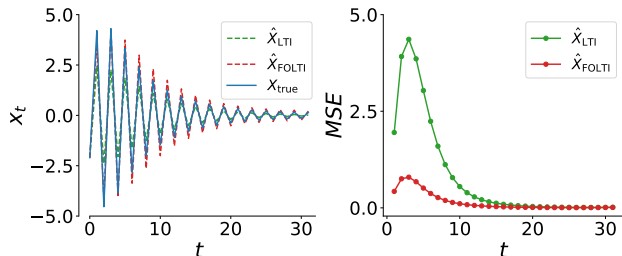

*Figure 3.* Numerical simulation of theoretical approaches for learning fractional-order dynamics.

***FOLOC framework evaluation.*** We evaluate our proposed data-driven approach by varying different variables. Our main findings are as follows: (i) FOLOC framework effectively captures the underlying system dynamics with high accuracy and remains robust to variations in fractional orders, system dimensions, noise types, noise scales, and training samples. (ii) The model's ability to learn the system dynamics improves as the time horizon increases. (iii) FOLOC framework can be trained using a small number of samples.

**Varying fractional orders.** By varying the fractional order $\alpha$, we observe that FOLOC framework effectively captures long-term dependencies. The results of our experiments on commensurate FOLTI systems (see Appendix C) with different fractional orders $\alpha = 0.1, 0.3, 0.5, 0.7, 0.9$ are shown in Table 1. As the fractional order $\alpha$ increases, we observe that the MSE and MAE remain approximately the same, with an overall mean MSE of 0.40% (standard deviation: 0.21%) and an overall mean MAE of 0.95% (standard deviation: 0.55%).

**Varying time horizons.** We show the results of learning fractional-order dynamics with different time horizons $T = 8, 16, 32, 64, 128$ in Table 1. As the time horizon increases, FOLOC framework demonstrates improved ability to capture the fractional-order dynamics. Compared with at $T = 8$, the MSE and MAE at $T = 128$ of FOLOC framework decrease by 97.37% and 92.23%, respectively.

*Table 1.* Experimental results on synthetic FOLTI system under varying input conditions. All reported values are in units of $10^{-3}$. **Bolded** values represent the minimum in each row.

| Metrics | **MSE/MAE** | **MSE/MAE** | **MSE/MAE** | **MSE/MAE** | **MSE/MAE** |
|---|---|---|---|---|---|
| Fractional | alpha = 0.1 | alpha = 0.3 | alpha = 0.5 | alpha = 0.7 | alpha = 0.9 |
| Order | **3.39±0.52/8.04±0.42** | 5.04±1.82/12.5±1.11 | 5.61±0.84/16.5±0.61 | 5.18±0.06/9.22±0.08 | 4.94±0.09/10.5±0.09 |
| State | $n = 1$ | $n = 2$ | $n = 4$ | $n = 6$ | $n = 8$ |
| Dimension | **1.98±0.08/6.60±0.13** | 5.61±0.84/16.5±0.61 | 5.84±0.71/13.4±0.55 | 4.88±0.34/13.4±0.26 | 4.26±0.22/12.3±0.14 |
| Time | T = 8 | T = 16 | T = 32 | T = 64 | T = 128 |
| Horizon | 101±18.4/111±4.40 | 43.2±8.20/59.1±1.12 | 14.2±2.96/31.5±1.70 | 5.61±0.84/16.5±0.61 | **2.66±0.64/8.62±0.70** |
| Noise Type | Cauchy | Gamma | Sinc-squared | Uniform | Poisson |
| | 6.98±1.89/17.0±0.71 | **5.84±1.15**/16.7±1.16 | 6.87±1.46/17.6±0.75 | 6.46±1.29/17.0±1.30 | 6.59±1.59/**16.5±0.90** |
| Noise Scale | $\sigma = 0.0001$ | $\sigma = 0.001$ | $\sigma = 0.01$ | $\sigma = 0.1$ | $\sigma = 1$ |
| (Gaussian) | 5.85±0.65/16.9±0.65 | 6.09±0.74/17.4±0.87 | 5.74±0.83/16.7±0.86 | **5.65±0.70/16.0±0.39** | 10.2±8.11/20.5±7.37 |
| Training Size | $N_T = 1000$ | $N_T = 2000$ | $N_T = 4000$ | $N_T = 6000$ | $N_T = 8000$ |
| | 5.18±1.13/12.9±0.80 | 4.44±1.30/11.4±0.38 | **3.98±0.89**/10.9±1.01 | 4.79±2.11/10.4±0.58 | 5.22±2.31/**10.1±0.78** |

**Varying state dimensions.** We vary the system state dimension $n = 1, 2, 4, 6, 8$ to evaluate its impact on the learning process (assume $n = m$). Increasing the state dimension significantly raises the learning complexity due to the larger number of variables and the interactions or couplings between the states. Despite this, FOLOC framework demonstrates consistent performance, with an overall mean MSE of 0.45% (standard deviation: 0.15%) and an overall mean MAE of 1.24% (standard deviation: 0.36%). Compared with at $n = 2$, the MSE (MAE) of FOLOC framework decreases by 13.01% (18.79%) at $n = 6$ and 24.06% (25.45%) at $n = 8$. This experiment demonstrates the scalability of our model on moderate-dimensional systems. We further evaluate the scalability of the FOLOC framework in high-dimensional settings ($n = m$), as shown in Table 2. The results verify its ability to capture interactions within the fractional dynamics of the state as the system dimensionality increases.

*Table 2.* Experiments on high-dimensional systems.

| Metrics | MSE/MAE |
|---|---|
| Dimension 16 | $3.85 \pm 0.05/12.7 \pm 0.08$ |
| Dimension 32 | $2.8 \pm 0.04/11.0 \pm 0.07$ |
| Dimension 64 | $1.8 \pm 0.001/8.39 \pm 0.007$ |
| Dimension 128 | $1.43 \pm 0.01/7.27 \pm 0.03$ |

**Varying training samples.** We show the model performance with different number of training samples $N = 1000, 2000, 4000, 6000, 8000$. Compared with at $N = 8000$, the MSE only decreases by 1.65% at $N = 1000$. This demonstrates that FOLOC framework is sample-efficient and capable of learning the underlying fractional-order dynamics with a limited number of samples.

**Varying noise distributions.** By varying the distribution of

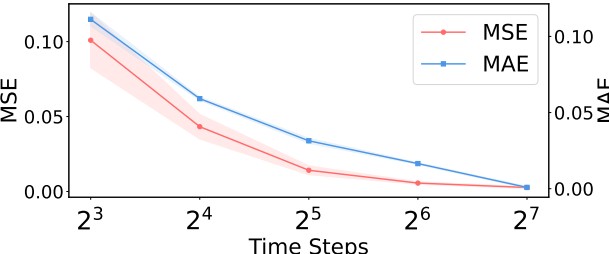

*Figure 4.* Vary time horizons.

process noise, we demonstrate the robustness and effectiveness of FOLOC framework. We evaluate the model's performance by simulating the noise from the following probability distribution functions: Gaussian, Poisson, Uniform, Gamma, Sinc-squared (Adigun & Kosko, 2023; Zhang et al., 2024), and Cauchy. Compared with using Gaussian noise, the maximal increases of MSE/MAE by 24.42%/6.67% (Cauchy/Sinc-squared). The empirical results (Table 1) show that FOLOC framework can not only predict an approximately accurate optimal control sequence but also effectively estimate the system parameters.

**Varying noise scales.** FOLOC framework shows the ability to learning the fractional-order dynamics with data corrupted by different Gaussian noise scales. As data is corrupted by different scales of Gaussian noise, FOLOC framework demonstrates consistent performance, with an overall mean MSE of 0.67% (standard deviation: 0.19%) and an overall mean MAE of 1.75% (standard deviation: 0.18%).

The additional results also demonstrate that the FOLOC framework is computationally efficient (see Appendix J.3) and capable of learning from multiple FOLTI systems (see Apendix J.2). We also evaluate the performance by varying the weight parameter $\lambda_w$ in the loss function (28) (see

Appendix H.2).

## 4.2. Performance Evaluation on Real-World Systems

To demonstrate that our deep learning model can effectively capture complex system dynamics, we evaluate it on two practical scenarios: cart-pole dynamics and quadrotor dynamics (Zhou & Tzoumas, 2024). The details of the system dynamics are given in the Appendix E.3.

**Cart Pole.** The empirical experiments show that FOLOC framework can correctly capture the underlying system dynamics even when we add some large-scale Gaussian noise. The overall mean MSE is less than $3.99 \times 10^{-5}$ and the overal mean MAE is less than $3.39 \times 10^{-3}$.

*Table 3.* Model performance in the cart pole experiment.

| Metrics | Noise Scale ($\sigma$) | | | | No noise |
|---|---|---|---|---|---|
| | 1 | $10^{-1}$ | $10^{-2}$ | $10^{-3}$ | |
| MSE ($10^{-5}$) | 4.0 | 3.9 | 4.1 | 3.9 | 3.9 |
| MAE ($10^{-3}$) | 3.4 | 3.4 | 3.4 | 3.3 | 3.3 |

*Table 4.* Baseline Model Evaluation on cart-pole dynamics. **Bolded** values represent the minimum in each column.

| Metrics | MSE/MAE | Run Time (s) |
|---|---|---|
| MBRL | 0.195/0.411 | 0.045 |
| CMA-ES | 0.237/0.472 | 16.405 |
| MPC | 0.237/0.473 | 0.674 |
| PID | 0.254/0.499 | 0.001 |
| iLQR | 0.238/0.474 | 0.263 |
| **FOLOC** | **0.00004/0.0034** | **0.0004** |

**Quadrotor.** The empirical experiments show that FOLOC framework is robust to different Gaussian noise scales when learning the Quadrotor dynamics. The overall mean MSE is 0.13 (standard deviation: 0.052), and the overall mean MAE is 0.21 (standard deviation: 0.086).

*Table 5.* Model performance in the quadrotor experiment.

| Metrics | Noise Scale ($\sigma$) | | | | No noise |
|---|---|---|---|---|---|
| | 1 | $10^{-1}$ | $10^{-2}$ | $10^{-3}$ | |
| MSE ($10^{-1}$) | 1.27 | 1.27 | 1.25 | 1.32 | 1.23 |
| MAE ($10^{-1}$) | 2.10 | 2.14 | 2.08 | 2.13 | 2.06 |

**Baseline Comparisons.** We compare the proposed FOLOC framework against several existing control methods, including model-based reinforcement learning (MBRL), covariance matrix adaptation evolution strategy (CMA-ES), model predictive control (MPC), proportional–integral–derivative (PID) control, and the iterative linear quadratic regulator (iLQR), evaluated on both cart-pole and quadrotor dynamics. For each method, we report predictive performance and runtime. FOLOC framework consistently achieves the

*Table 6.* Baseline Model Evaluation on quadrotor dynamics. **Bolded** values represent the minimum in each column.

| Metrics | MSE/MAE | Run Time (s) |
|---|---|---|
| MBRL | 0.489/0.546 | 0.385 |
| CMA-ES | 0.613/0.590 | 115.151 |
| MPC | 0.612/0.552 | 71.943 |
| PID | 1.143/0.909 | 0.0053 |
| iLQR | 0.628/0.570 | 3.387 |
| **FOLOC** | **0.211/0.127** | **0.00027** |

highest accuracy and the lowest computational cost across both tasks. Notably, FOLOC framework reduces the mean squared error (MSE) by up to 99.99% and the mean absolute error (MAE) by up to 99.32% compared to existing baselines on the cart-pole dynamics task. All experimental code is available at the link provided in Appendix A.

**Remark.** The FOLOC framework, an end-to-end learning-based LQR in FOLTI systems, offers several key advantages: (i) FOLOC framework generalizes across different FOLTI systems (see Appendix J.2), predicting system parameters and optimal control policies directly from trajectories. Notably, system parameters are not used during testing on synthetic data, and are not used entirely during both training and testing on real-world dynamics. (ii) FOLOC framework enables fast and accurate real-time control. While closed-form solutions (Eqs. 16 and 17) are available given full system knowledge, they require large-scale matrix inversions and become computationally infeasible for long horizons or high-dimensional systems. MPC, though an alternative, also struggles with the memory effects in fractional-order dynamics. FOLOC framework addresses these challenges by learning a direct mapping from observed trajectories and cost matrices to optimal control solutions.

## 5. Conclusion

This paper introduces a theoretical learning framework for optimal control of FOLTI systems, addressing key challenges in system identification, control design, and sample complexity analysis. By extending LQR to the fractional-order setting, we derive analytical solutions and provide sample complexity guarantees. Based on this theoretical foundation, we propose FOLOC, an end-to-end framework that jointly estimates system parameters and learns optimal control policies from noisy trajectories. Unlike classical methods, FOLOC framework is designed to operate effectively in real-world, noisy environments, ensuring robustness against non-Gaussian noise and limited training samples. Extensive simulations and real-world applications demonstrate the scalability, robustness, and computational efficiency of our approach.

## Acknowledgements

The authors acknowledge the support by the U.S. Army Research Office (ARO) under Grant No. W911NF-23-1-0111, the National Science Foundation (NSF) under the Career Award CPS-1453860, Career Award 2337412, MCB-1936775, CNS-1932620 and the NSF Award 2243104 under the Center for Complex Particle Systems (COMPASS), the Defense Advanced Research Projects Agency (DARPA) Young Faculty Award and DARPA Director Fellowship Award under Grant Number N66001-17-1-4044, Intel faculty awards and a Northrop Grumman grant. P.B. is also grateful to National Institute of Health (NIH) for the grants R01 AG 079957 "Interpretable machine learning to synergize brain age estimation and neuroimaging genetics" and RF1 AG 082201 "Neurovascular calcification and ADRD in two nonindustrial Native American populations". It was a wonderful experience designing and writing the grant application entitled "Neurovascular calcification and ADRD in two nonindustrial Native American populations" and awarded under RF1 AG 082201. The views, opinions, and/or findings in this article are those of the authors and should not be interpreted as official views or policies of the Department of Defense, the National Institute of Health or the National Science Foundation. The authors would like to thank Zehao Zhao and Zonglin Zhang for their insightful discussions during the course of this project.

## Impact Statement

Fractional-order dynamical systems can model complex dynamical behaviors characterized by long-range dependencies and memory effects, yet their practical implementation remains challenging due to the lack of system identification and control strategies. This work advances the field by formulating both theoretical and deep learning frameworks for solving fractional-order optimal control via the LQR. For theoretical learning, we provide a new fractional-order system identification and a new sample complexity analysis that gives guarantees on the number of samples required for achieving reliable control performance. Beyond theoretical contributions, we develop a novel end-to-end data-driven learning framework that jointly estimates system parameters and optimizes control policies directly from observed trajectories. Unlike traditional methods that rely on strong assumptions (e.g., noiseless data or structured disturbances), our approach remains robust under realistic, noisy environments with non-Gaussian uncertainties.

By bridging theoretical foundations with deep learning techniques, our work enables scalable and efficient optimal control for fractional-order dynamical systems in domains such as biomedical engineering, neuroscience, and financial modeling. Experimental results validate the effectiveness of our approach, demonstrating accurate system identification and control policy learning across various noise distributions and system complexities.

This research lays the groundwork for future advancements in learning-based control of fractional-order dynamical systems, paving the way for real-world deployment of fractional-order optimal controllers in autonomous systems, robotics, and large-scale networks.

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

**Limitation**: While our framework is effective for FOLTI systems, its LTI-based design requires further validation in non-LTI settings. Future work will extend the framework to non-LTI dynamics, enhancing its applicability to more complex systems.

## A. Code Availability

The source code is available at https://github.com/zpykillcc/Fractional-Order-Learning-for-Optimal-Control-Framework.

## B. Related Work

While research on system identification and optimal control for FOLTI systems remains limited, significant progress has been made in LTI systems. Estimating unknown parameters of linear dynamical systems is a well-established subfield of system identification in control theory (Sarkar et al., 2021; Faradonbeh et al., 2018; Simchowitz et al., 2018), where the task is to estimate parameters from input-output time series generated by the underlying system. With the increasing interest in machine learning and deep learning, many researchers use different deep learning based modeling techniques to solve general nonlinear system identification. Chen et al. (1990) shows a single hidden layer neural network can identify discrete time nonlinear systems and derives new parameter estimation algorithms based on a prediction error formulation. Narendra & Parthasarathy (1990) demonstrates that neural networks can be used effectively for both identification and control of nonlinear dynamical systems. The hierarchical structures of multilayer feedforward neural networks can include dynamic systems features (Zancato & Chiuso, 2021) and bring extra flexibility for probablistic approaches (Hendriks et al., 2020). Kernel-based methods also have been studied in system identification. Linear system identification can be seen as an application of learning the impulse response function, which can be approximated by kernels (Aronszajn, 1950; Dinuzzo & Schölkopf, 2012; Cho & Saul, 2009). Deep state-space models (Gedon et al., 2021) like recurrent neural networks (Hochreiter, 1997; Cho et al., 2014) and autoencoders (Masti & Bemporad, 2021; Lusch et al., 2018) are also gaining in popularity for system identification. For Another critical aspect of optimal control via LQR is the design of control inputs, which are typically formulated as linear combinations of disturbance processes (Dean et al., 2020).

Solving the optimal control problem via LQR involves addressing a potentially large linear system with a block Toeplitz matrix of size $\mathbb{R}^{Tn \times Tn}$. While solutions for Toeplitz and block Toeplitz systems have been extensively studied (Trench, 1986; Kalouptsidis et al., 1984; Chandrasekaran & Sayed, 1998), they still present significant computational challenges. Direct solution methods for Toeplitz systems typically exhibit a computational complexity of $\mathcal{O}(N^2)$, where $N$ represents the degrees of freedom. However, the high memory requirements of these methods often limit their applicability to large-scale problems. Iterative solvers (Chan & Jin, 2007; Strang, 1986) are more memory-efficient and can achieve a complexity of $\mathcal{O}(N \log^2(N))$ for a single linear block Toeplitz system. Advanced approaches, such as global and block variants of the generalized minimal residual (GMRES) method (Saad & Schultz, 1986), can efficiently address sequences of block Toeplitz systems with multiple right-hand sides (Jelich et al., 2021). Machine learning-based methods, such as neural operators (Li et al., 2021; Gupta et al., 2021; 2022; Xiao et al., 2022), offer a promising alternative for accelerating the solution of linear systems. Neural operator-assisted Krylov iterations (Luo et al., 2024), for example, have demonstrated significant computational advantages, achieving up to a $5.5\times$ speedup in computation time and a $16.1\times$ reduction in the number of iterations. These advancements underscore the potential of machine learning techniques to address computational bottlenecks in optimal control for FOLTI systems.

## C. Why Fractional Order?

According to the well-known definition, the first-order derivative of the function $f(t)$, denoted by $\mathcal{D}^1 f(t)$, is defined by

$$\mathcal{D}^1 f(t) = \frac{df(t)}{dt} = \lim_{h \to 0} \frac{f(t) - f(t-h)}{h}, \tag{31}$$

that is, as the limit of a backward difference. Similarly,

$$\mathcal{D}^2 f(t) = \frac{d^2 f(t)}{dt^2} = \lim_{h \to 0} \frac{1}{h^2} \big[ f(t) - 2f(t-h) + f(t-2h) \big] \tag{32}$$

and

$$\mathcal{D}^3 f(t) = \frac{d^3 f(t)}{dt^3} = \lim_{h \to 0} \frac{1}{h^3} \big[ f(t) - 3f(t-h) + 3f(t-2h) - f(t-3h) \big]. \tag{33}$$

Iterating $n$-times, we can obtain

$$\mathcal{D}^n f(t) = \frac{d^n f(t)}{dt^n} = \lim_{h \to 0} \frac{1}{h^n} \sum_{k=0}^{n} (-1)^n \binom{n}{k} f(t - kh), \tag{34}$$

where

$$\binom{n}{k} = \frac{n(n-1)(n-2)\cdots(n-k+1)}{k!}. \tag{35}$$

Naturally we can extend the common binomial coefficients for any $n \in \mathbb{R}^+$ by letting $(n-1)! = \Gamma(n)$. So the Riemann-Liouville fractional-order integral can be defined as a consequence of Cauchy's formula for repeated integrals.

**Definition C.1** (Riemann-Liouville fractional-order integral). Given a positive real number $\alpha$, the Riemann-Liouville fractional-order integral is defined as

$$\mathcal{I}_c^\alpha f(t) := \frac{1}{\Gamma(\alpha)} \int_c^t (t - \tau)^{\alpha - 1} f(\tau) \, d\tau, \quad t > c, \, \alpha \in \mathbb{R}^+. \tag{36}$$

Consider an integer-order derivative operator $\mathcal{D}^n$ ($n \in \mathbb{N}$). The derivative operator $\mathcal{D}^n$ is only a left-inverse of the integral operator $\mathcal{I}^n$ such that $\mathcal{D}^n \mathcal{I}^n = \mathbb{I}$ and $\mathcal{I}^n \mathcal{D}^n \neq \mathbb{I}$. In order to generalize integer-order derivatives, the fractional-order derivative operator $\mathcal{D}^\alpha$ should follow the same left-inverse rule. We show the definition of the Riemann-Liouville fractional-order derivative as follows.

**Definition C.2** (Riemann–Liouville fractional-order derivative). Let $f : \mathbb{R} \to \mathbb{R}$ be a function and let $I = [a, b]$ be a finite interval on the real axis $\mathbb{R}$. The left-side and right-side fractional-order derivatives ${}^{RL}D_{a+}^\alpha f$ and ${}^{RL}D_{b-}^\alpha f$ of order $\alpha \in \mathbb{R}$ are defined by

$$\left( {}^{RL}\mathcal{D}_{a+}^\alpha f \right)(x) = \frac{1}{\Gamma(n-\alpha)} \left( \frac{d}{dx} \right)^n \int_a^x \frac{f(t)\, dt}{(x-t)^{\alpha - n + 1}}, \tag{37}$$

$$\left( {}^{RL}\mathcal{D}_{b-}^\alpha f \right)(x) = \frac{1}{\Gamma(n-\alpha)} \left( -\frac{d}{dx} \right)^n \int_x^b \frac{f(t)\, dt}{(t-x)^{\alpha - n + 1}}, \tag{38}$$

where $n - 1 \leq \alpha \leq n$, $x > a$ in (37), and $x < b$ in (38).

The Grünwald–Letnikov fractional-order derivative is defined as the limit of finite differences and is mathematically equivalent to the Riemann–Liouville fractional-order derivative. The Grünwald–Letnikov fractional-order derivative plays an important role establishing fractional-order dynamical systems. The equations for a continuous-time fractional-order dynamical system can be written as follows:

$$H \left( \mathcal{D}^{\alpha_0 \alpha_1 \alpha_2 \cdots \alpha_n} \right) (y_1, y_2, \cdots, y_l) = G \left( \mathcal{D}^{\beta_0 \beta_1 \beta_2 \cdots \beta_m} \right) (u_1, u_2, \cdots, u_k), \tag{39}$$

where $y_i, u_i$ are functions of time and $H(\cdot), G(\cdot)$ are the combination laws of the fractional-order derivative operator.

A single-input single-output LTI fractional-order system can be described by a fractional differential equation of the form

$$\sum_{k=0}^{n} a_k \mathcal{D}^{\alpha_k} y(t) = \sum_{k=0}^{m} b_k \mathcal{D}^{\beta_k} u(t), \tag{40}$$

with a corresponding transfer function of the form:

$$G(s) = \frac{Y(s)}{U(s)} = \frac{\sum_{k=0}^{m} b_k s^{\beta_k}}{\sum_{k=0}^{n} a_k s^{\alpha_k}}, \tag{41}$$

where $a_k, b_k \in \mathbb{R}$. The system is said to be commensurate if $\alpha_k = \beta_k = k\alpha$, where $\alpha \in \mathbb{R}^+$, otherwise the system is non-commensurate.

We can use the Grünwald–Letnikov fractional-order derivative to rewrite the fractional-order differential equation as follows in the discrete-time case:

$$\sum_{k=0}^{n} a_k \Delta_h^{\alpha_k} y(t) = \sum_{k=0}^{m} b_k \Delta_h^{\beta_k} u(t), \tag{42}$$

with a corresponding transfer function of the following form:

$$G(z) = \frac{\sum_{k=0}^{m} b_k \left( \omega(z^{-1}) \right)^{\beta_k}}{\sum_{k=0}^{n} a_k \left( \omega(z^{-1}) \right)^{\alpha_k}},$$  (43)

where $\omega(z^{-1})$ is the $Z$ transform of the operator $\Delta_h^1$.

## D. Theorems and Proofs

### D.1. System Identification

We aim to estimate the parameters $\Theta = \{A, B, \alpha\}$ of the following FOLTI system:

$$x_{k+1} = Ax_k + Bu_k - \sum_{j=1}^{k+1} \psi(\alpha, j)x_{k+1-j}.$$  (44)

To note, we can not identify $\Theta$ jointly so we make some assumptions during the system identification.

#### D.1.1. DATA GENERATION

In this step, we generate data following the procedure outlined below:

$$
\begin{aligned}
x_1^1 &= Ax_0^1 + Bu_0^1 - \psi(\alpha, 1)x_0^1 + w_0^1 \\
&= Ax_0^1 + Bu_0^1 + C_\alpha x_0^1 + w_0^1 \\
x_1^2 &= Ax_0^2 + Bu_0^2 - \psi(\alpha, 1)x_0^2 + w_0^2 \\
&= Ax_0^2 + Bu_0^2 + C_\alpha x_0^2 + w_0^2
\end{aligned}
$$  (45)
(46)

$$
\begin{aligned}
&\ldots \\
x_1^p &= Ax_0^p + Bu_0^p - \psi(\alpha, 1)x_0^p + w_0^p \\
&= Ax_0^p + Bu_0^p + C_\alpha x_0^p + w_0^p
\end{aligned}
$$  (47)

where $x_0^i$ are the $i$-th initial conditions, $x_1^i$ represents the corresponding state at the second time step generated from the initial condition, and $w_0^i$ are independent and identically distributed (i.i.d.) white Gaussian noise for $i = 1, \ldots, p$, and $C_\alpha = \text{diag}(\alpha_1, \alpha_2, \ldots, \alpha_n)$.

#### D.1.2. ESTIMATION

We can write (86) as follows

$$
\begin{aligned}
x_1^i &= Ax_0^i + Bu_0^i + C_\alpha x_0^i + w_0^i \\
&= \underbrace{\begin{bmatrix} \gamma_1^1 + \alpha_1 & \gamma_2^1 & \cdots & \gamma_n^1 \\ \gamma_1^2 & \gamma_2^2 + \alpha_2 & \cdots & \gamma_n^2 \\ \vdots & \vdots & \cdots & \vdots \\ \gamma_1^n & \gamma_2^n & \cdots & \gamma_n^n + \alpha_n \end{bmatrix}}_{A_\alpha} x_0^i + \underbrace{\begin{bmatrix} \beta_1^1 & \beta_2^1 & \cdots & \beta_m^1 \\ \beta_1^2 & \beta_2^2 & \cdots & \beta_m^2 \\ \vdots & \vdots & \cdots & \vdots \\ \beta_1^n & \beta_2^n & \cdots & \beta_m^n \end{bmatrix}}_{B} u_0^i + w_0^i \\
&= \underbrace{\begin{bmatrix} {x_0^i}^\top & 0 & \cdots & \cdots \\ 0 & {x_0^i}^\top & 0 & \cdots \\ \vdots & \vdots & \ddots & \vdots \\ 0 & \cdots & \cdots & {x_0^i}^\top \end{bmatrix}}_{\pi_i} \begin{bmatrix} \gamma_1 \\ \vdots \\ \gamma_n \end{bmatrix} + \underbrace{\begin{bmatrix} {u_0^i}^\top & 0 & \cdots & \cdots \\ 0 & {u_0^i}^\top & 0 & \cdots \\ \vdots & \vdots & \ddots & \vdots \\ 0 & \cdots & \cdots & {u_0^i}^\top \end{bmatrix}}_{\phi_i} \begin{bmatrix} \beta_1 \\ \vdots \\ \beta_n \end{bmatrix} + w_0^i
\end{aligned}
$$  (48)

Thus, given the generated data, we solve the following least squares problem:

$$\underbrace{\begin{bmatrix} x_1^1 \\ \vdots \\ x_1^p \end{bmatrix}}_{X} = \underbrace{\begin{bmatrix} \pi_1 \\ \vdots \\ \pi_p \end{bmatrix}}_{\pi} \underbrace{\begin{bmatrix} \gamma_1 \\ \vdots \\ \gamma_n \end{bmatrix}}_{\gamma} + \underbrace{\begin{bmatrix} \phi_1 \\ \vdots \\ \phi_p \end{bmatrix}}_{\phi} \underbrace{\begin{bmatrix} \beta_1 \\ \vdots \\ \beta_n \end{bmatrix}}_{\beta} + \underbrace{\begin{bmatrix} w_0^1 \\ \vdots \\ w_0^p \end{bmatrix}}_{\omega}, \tag{49}$$

$$X = \pi\gamma + \phi\beta + w = \begin{bmatrix} \pi & \phi \end{bmatrix} \begin{bmatrix} \gamma \\ \beta \end{bmatrix} + w. \tag{50}$$

Then we have the following optimization problem

$$\hat{\theta} = \arg\min_{\theta} \|X - \xi\theta\|_2^2, \tag{51}$$

with the solution

$$\hat{\theta} = \left(\xi^\top \xi\right)^{-1} \xi^\top X, \tag{52}$$

where $\theta = [\gamma^\top, \beta^\top]^\top$, and $\xi = [\pi, \phi]$.

Follow (Zhang et al., 2025), we can revise the above strategy and formulate the sample complexity problem as follows:

$$\begin{aligned} \theta_{t+1} &= \theta_t, \\ X_t &= \xi\theta_t + w_t, \end{aligned} \tag{53}$$

where the state vector $\theta_t = [\gamma^\top, \beta^\top]^\top$, and $w_k$ is a white Gaussian noise vector $w_t \sim \mathcal{N}(0, K_w)$, where $K_w$ is a diagonal covariance matrix, and $t = 1, 2, \ldots, N$. Thus, the solution $\hat{\theta}$ has the following Gaussian distribution:

$$\hat{\theta} \sim N\left(\theta, \frac{1}{N}(\xi\top\xi)^{-1}\xi\top K_w \xi(\xi\top\xi)^{-1}\right). \tag{54}$$

Notice that the matrix $A$ and the fractional order $\alpha$ are coupled. Therefore, if the diagonal elements of $A$ are known, the fractional order $\alpha$ can be identified by subtracting these constant values from the corresponding positions in the estimated parameter vector $\hat{\theta}$.

### D.2. Proof of Lemma 3.1

*Proof.* By mathematical induction as follows:

(1) When $t = 0, x[0] = G_0 x[0] = x[0]$. When $t = 1, x[1] = G_1 x[0] + Bu[0]$

(2) Assume we have $x[k] = G_k x[0] + \sum_{j=0}^{k-1} G_{k-1-j} Bu[j]$, when $t = k$.

When $t = k + 1$,

$$\begin{aligned} x[k+1] &= \sum_{j=0}^{k} A_j x[k-j] + Bu[k] \\ &= \sum_{j=0}^{k} (A_j (G_{k-j} x[0] + \sum_{i=0}^{k-j-1} G_{k-1-i-j} Bu[i])) + Bu[k] \\ &= \sum_{j=0}^{k} A_j G_{k-j} x[0] + \sum_{j=0}^{k} A_j (\sum_{i=0}^{k-j-1} G_{k-1-i-j} Bu[i]) + Bu[k] \\ &= G_{k+1} x[0] + \sum_{j=0}^{k} \sum_{i=0}^{k-j-1} A_j (G_{k-1-i-j} Bu[i]) + Bu[k] \end{aligned} \tag{55}$$

By expanding the second term, thus

$$x[k+1] = G_{k+1} x[0] + \sum_{j=0}^{k} G_{k-j} Bu[j]. \tag{56}$$

$\square$

## D.3. Proof of Theorem 3.2

*Proof of least-squares solution.* Rewrite as follows:

$$
\begin{bmatrix} x_0 \\ x_1 \\ x_2 \\ x_3 \\ \vdots \\ x_T \end{bmatrix} = \underbrace{\begin{bmatrix} 0 & 0 & 0 & 0 & \cdots & 0 \\ G_0 B & 0 & 0 & 0 & \cdots & 0 \\ G_1 B & G_0 B & 0 & 0 & \cdots & 0 \\ G_2 B & G_1 B & G_0 B & 0 & \cdots & 0 \\ \vdots & \vdots & \vdots & \vdots & \ddots & \vdots \\ G_{T-1} B & G_{T-2} B & G_{T-3} B & G_{T-4} B & \cdots & G_0 B \end{bmatrix}}_{G} \begin{bmatrix} u_0 \\ u_1 \\ u_2 \\ u_3 \\ \vdots \\ u_{T-1} \end{bmatrix} + \underbrace{\begin{bmatrix} I \\ G_1 \\ G_2 \\ G_3 \\ \vdots \\ G_T \end{bmatrix}}_{H} x_0. \tag{57}
$$

$$
X = GU + Hx_0 \tag{58}
$$

Rewrite as follows:

$$
J_T(U) = \min_U \left\{ X^\top \bar{Q} X + U^\top \bar{R} U \right\} \tag{59}
$$

where

$$
\bar{Q} = \begin{bmatrix} Q & 0 & 0 & 0 & \cdots & 0 \\ 0 & Q & 0 & 0 & \cdots & 0 \\ 0 & 0 & Q & 0 & \cdots & 0 \\ 0 & 0 & 0 & Q & \cdots & 0 \\ \vdots & \vdots & \vdots & \vdots & \ddots & \vdots \\ 0 & 0 & 0 & 0 & \cdots & Q_f \end{bmatrix}
$$

$$
\bar{R} = \begin{bmatrix} R & 0 & 0 & 0 & \cdots & 0 \\ 0 & R & 0 & 0 & \cdots & 0 \\ 0 & 0 & R & 0 & \cdots & 0 \\ 0 & 0 & 0 & R & \cdots & 0 \\ \vdots & \vdots & \vdots & \vdots & \ddots & \vdots \\ 0 & 0 & 0 & 0 & \cdots & R \end{bmatrix} \tag{60}
$$

Substitute (59) using (58):

$$
\min_U J_T(U) = \min_U \left\{ (GU + Hx_0)^\top \bar{Q} (GU + Hx_0) + U^\top \bar{R} U \right\} \tag{61}
$$

The solution for the least squares problem is

$$
(G^\top \bar{Q} G + \bar{R}) U = -G^\top \bar{Q}^\top H x_0
$$
$$
U = -(G^\top \bar{Q} G + \bar{R})^{-1} G^\top \bar{Q}^\top H x_0 \tag{62}
$$

$\square$

*Proof of Lagrange multiplier solution.*

**Lemma D.1** (Lagrange multiplier condition). *The Lagrangian function $\mathcal{L}$ is given by:*

$$
\mathcal{L}(x, u, \lambda) = \sum_{k=0}^{T-1} \left( x_k^\top Q x_k + u_k^\top R u_k + \lambda_{k+1}^\top (A x_k + B u_k - \Delta^\alpha x_{k+1}) \right) + x_T^\top Q_f x_T. \tag{63}
$$

*and the solution is given by:*

$$
u_k = -\frac{1}{2} R^{-1} B^\top \lambda_{k+1},
$$

$$
\lambda_k = 2 Q x_k + A^\top \lambda_{k+1} - \sum_{i=k+1}^{T} D(\alpha, i - k) \lambda_i \tag{64}
$$

with $x_0 = x_0$ and $\lambda_T = 2Q_f x_T$.

*Proof of Lemma 1.* Taking the partial derivative with respect to $u_k, x_k, \lambda_k$, we have

$$\frac{\partial \mathcal{L}}{\partial u_k} = 2Ru_k + B^\top \lambda_{k+1} = 0 \implies u_k = -\frac{1}{2}R^{-1}B^\top \lambda_{k+1}, \tag{65}$$

$$\frac{\partial \mathcal{L}}{\partial x_k} = 2Qx_k + A^\top \lambda_{k+1} - \sum_{i=k}^{T} D(\alpha, i - k)\lambda_i = 0, \tag{66}$$

$$\implies \lambda_k = 2Qx_k + A^\top \lambda_{k+1} - \sum_{i=k+1}^{T} D(\alpha, i - k)\lambda_i \tag{67}$$

$$\frac{\partial \mathcal{L}}{\partial \lambda_k} = Ax_k + Bu_k - \Delta^\alpha x_{k+1} = 0. \tag{68}$$

Explanation for Eq. (66): From Eq. (3), we have

$$\Delta^\alpha x_k = \sum_{j=0}^{k} D(\alpha, j)x[k - j] \tag{69}$$

Then, we have

$$\frac{\partial}{\partial x_k}\left(\lambda_k^\top \Delta^\alpha x_k\right) = D(\alpha, 0)\lambda_k$$

$$\frac{\partial}{\partial x_k}\left(\lambda_{k+1}^\top \Delta^\alpha x_{k+1}\right) = D(\alpha, 1)\lambda_{k+1}$$

$$\vdots$$

$$\frac{\partial}{\partial x_k}\left(\lambda_T^\top \Delta^\alpha x_T\right) = D(\alpha, T - k)\lambda_T \tag{70}$$

$\square$

Solve the $\lambda$ jointly and then substitute in (65) to get $u_k$.

Note we have the following equations by (56), (65) and (66):

$$u_k = -\frac{1}{2}R^{-1}B^\top \lambda_{k+1},$$

$$\lambda_k = 2Qx_k + A^\top \lambda_{k+1} - \sum_{i=k+1}^{T} D(\alpha, i - k)\lambda_i$$

$$x[k] = G_k x[0] + \sum_{j=0}^{k-1} G_{k-1-j}Bu[j] \tag{71}$$

Obeserve that

$$\lambda_k = \begin{bmatrix} 2QG_k & -QG_{k-1}BR^{-1}B^\top & \cdots & -QG_0BR^{-1}B^\top & A^\top - D(\alpha, 1) & \cdots & -D(\alpha, T - k) \end{bmatrix} \begin{bmatrix} x_0 \\ \lambda_1 \\ \vdots \\ \lambda_k \\ \lambda_{k+1} \\ \vdots \\ \lambda_T \end{bmatrix} \tag{72}$$

We have the following matrix form, where $G_\lambda$ and $I - G_\lambda$ are block Toeplitz matrices (see Definition D.2).

$$
\begin{bmatrix} \lambda_1 \\ \lambda_2 \\ \lambda_3 \\ \vdots \\ \lambda_T \end{bmatrix} = \underbrace{\begin{bmatrix} -QG_0BR^{-1}B^\top & A^\top - D(\alpha,1) & -D(\alpha,2) & \cdots & -D(\alpha,T-1) \\ -QG_1BR^{-1}B^\top & -QG_0BR^{-1}B^\top & A^\top - D(\alpha,1) & \cdots & -D(\alpha,T-2) \\ -QG_2BR^{-1}B^\top & -QG_1BR^{-1}B^\top & -QG_0BR^{-1}B^\top & \cdots & -D(\alpha,T-3) \\ \vdots & \vdots & \vdots & \vdots & \ddots \\ -QG_{T-1}BR^{-1}B^\top & -QG_{T-2}BR^{-1}B^\top & \cdots & \cdots & -QG_0BR^{-1}B^\top \end{bmatrix}}_{G_\lambda} \begin{bmatrix} \lambda_1 \\ \lambda_2 \\ \lambda_3 \\ \vdots \\ \lambda_T \end{bmatrix}
$$

$$
+ 2 \underbrace{\begin{bmatrix} QG_1 \\ QG_2 \\ QG_3 \\ \vdots \\ QG_T \end{bmatrix}}_{H_\lambda} x_0. \tag{73}
$$

$$
\lambda = G_\lambda \lambda + 2H_\lambda x_0
$$
$$
(I - G_\lambda)\lambda = 2H_\lambda x_0
$$
$$
\lambda = 2(I - G_\lambda)^{-1} H_\lambda x_0 \tag{74}
$$

**Definition D.2** (Block Toeplitz matrix). A *block Toeplitz matrix* is a block matrix where the block structure follows a Toeplitz pattern. Specifically, let $\mathscr{T} \in \mathbb{R}^{mn \times mn}$ be partitioned into $m \times m$ blocks, each of size $n \times n$. The matrix $\mathscr{T}$ is defined as:

$$
\mathscr{T} = \begin{bmatrix} \mathscr{A}_0 & \mathscr{A}_{-1} & \cdots & \mathscr{A}_{-(m-1)} \\ \mathscr{A}_1 & \mathscr{A}_0 & \cdots & \mathscr{A}_{-(m-2)} \\ \vdots & \vdots & \ddots & \vdots \\ \mathscr{A}_{m-1} & \mathscr{A}_{m-2} & \cdots & \mathscr{A}_0 \end{bmatrix},
$$

where each $\mathscr{A}_k \in \mathbb{R}^{n \times n}$ represents a block matrix.

Using (65), we have

$$
U = -R^{-1}B^\top \otimes (I - G_\lambda)^{-1} H_\lambda x_0 \tag{75}
$$

$\square$

### D.4. Proof of Theorem 3.3

*Proof of least-squares sample complexity result.* If we know $A$ and $\alpha$, then $\theta$ will reduce to $vectorize(B)$ and the distribution reduces to the following:

$$
\hat{\theta} = vectorize(\hat{B}) \sim N\left(vectorize(B), \frac{1}{N}(\phi\top\phi)^{-1}\phi\top K_w \phi(\phi\top\phi)^{-1}\right). \tag{76}
$$

Then

$$
\mathbb{E}\left[|\hat{J} - J|\right] = \mathbb{E}\left[\|\hat{J} - J\|_2\right] = \mathbb{E}\left[\|\hat{a} - a\|_2\right]
$$

$$
= \mathbb{E}\left[\|z^\top \left(\underbrace{\hat{P}\left(\hat{P}^\top S\hat{P} + \bar{R}\right)^{-1}\hat{P}^\top}_{F(\hat{P})} - \underbrace{P\left(P^\top SP + \bar{R}\right)^{-1}P^\top}_{F(P)}\right)z\|_2\right]
$$

$$
\leq \mathbb{E}\left[\|F(\hat{P}) - F(P)\|_2\right]\|z\|_2^2 \tag{77}
$$

We rewrite $F(\hat{P}) - F(P)$ as follows:

$$
\begin{aligned}
F(\hat{P}) - F(P) &= \hat{P}(\hat{P}^\top S\hat{P} + \bar{R})^{-1}(\hat{P}^\top - P^\top) \\
&\quad + \left( \hat{P}(\hat{P}^\top S\hat{P} + \bar{R})^{-1} - P(P^\top SP + \bar{R})^{-1} \right) P^\top \\
&= \hat{P}\left( \hat{P}^\top S\hat{P} + \bar{R} \right)^{-1}\left( \hat{P}^\top - P^\top \right) \\
&\quad + \left( (\hat{P} - P)(\hat{P}^\top S\hat{P} + \bar{R})^{-1} + P\left( \left( \hat{P}^\top S\hat{P} + \bar{R} \right)^{-1} - \left( P^\top SP + \bar{R} \right)^{-1} \right) \right) P^\top \\
&= \hat{P}\left( \hat{P}^\top S\hat{P} + \bar{R} \right)^{-1}\left( \hat{P}^\top - P^\top \right) \\
&\quad + (\hat{P} - P)(\hat{P}^\top S\hat{P} + \bar{R})^{-1}P^\top + P\left( \left( \hat{P}^\top S\hat{P} + \bar{R} \right)^{-1} - \left( P^\top SP + \bar{R} \right)^{-1} \right) P^\top
\end{aligned}
\tag{78}
$$

Then we have

$$
\begin{aligned}
\mathbb{E}\left[ |\hat{J} - J| \right] &\leq \mathbb{E}\left[ \|F(\hat{P}) - F(P)\|_2 \right] \|z\|_2^2 \\
&\leq \underbrace{\mathbb{E}\left[ \left\| \hat{P}\left( \hat{P}^\top S\hat{P} + \bar{R} \right)^{-1}\left( \hat{P}^\top - P^\top \right) \right\|_2 \right]}_{(i)} \|z\|_2^2 \\
&\quad + \underbrace{\mathbb{E}\left[ \left\| (\hat{P} - P)(\hat{P}^\top S\hat{P} + \bar{R})^{-1}P^\top \right\|_2 \right]}_{(ii)} \|z\|_2^2 \\
&\quad + \underbrace{\mathbb{E}\left[ \left\| P\left( \left( \hat{P}^\top S\hat{P} + \bar{R} \right)^{-1} - \left( P^\top SP + \bar{R} \right)^{-1} \right) P^\top \right\|_2 \right]}_{(iii)} \|z\|_2^2
\end{aligned}
\tag{79}
$$

We bound term (i) as follows:

$$
\begin{aligned}
(i) \quad &\mathbb{E}\left[ \left\| \hat{P}\left( \hat{P}^\top S\hat{P} + \bar{R} \right)^{-1}\left( \hat{P}^\top - P^\top \right) \right\|_2 \right] \\
&\leq \mathbb{E}\left[ \left\| \hat{P} \right\|_2 \left\| \left( \hat{P}^\top S\hat{P} + \bar{R} \right)^{-1} \right\|_2 \left\| \hat{P} - P \right\|_2 \right] \\
&\leq \left\| \bar{R}^{-1} \right\|_2 \mathbb{E}\left[ \left\| \hat{P} \right\|_2 \left\| \hat{P} - P \right\|_2 \right].
\end{aligned}
\tag{80}
$$

We now use the following:

$$
\begin{cases}
\left\| \hat{P} \right\|_2 = \left\| \hat{P} - P + P \right\|_2 \leq \left\| \hat{P} - P \right\|_2 + \|P\|_2, \\
\left\| \hat{P} \right\|_2 \left\| \hat{P} - P \right\|_2 \leq \left( \left\| \hat{P} - P \right\|_2 + \|P\|_2 \right) \left\| \hat{P} - P \right\|_2 = \left\| \hat{P} - P \right\|_2^2 + \|P\|_2 \left\| \hat{P} - P \right\|_2.
\end{cases}
$$

$$
\begin{aligned}
&\leq \left\| \bar{R}^{-1} \right\|_2 \mathbb{E}\left[ \left\| \hat{P} - P \right\|_2^2 + \|P\|_2 \left\| \hat{P} - P \right\|_2 \right] \\
&= \left\| \bar{R}^{-1} \right\|_2 \mathbb{E}\left[ \left\| \hat{B} - B \right\|_2^2 + \|P\|_2 \left\| \hat{B} - B \right\|_2 \right] \\
&\leq \left\| \bar{R}^{-1} \right\|_2 \mathbb{E}\left[ \left\| \hat{B} - B \right\|_F^2 + \|P\|_2 \left\| \hat{B} - B \right\|_F \right] \\
&\leq \left\| \bar{R}^{-1} \right\|_2 \left( \mathrm{Tr}(K_B) + \|P\|_2 \sqrt{\mathrm{Tr}(K_B)} \right) \\
&= \left\| \bar{R}^{-1} \right\|_2 \left( \mathrm{Tr}(K_B) + \|B\|_2 \sqrt{\mathrm{Tr}(K_B)} \right).
\end{aligned}
\tag{81}
$$

We bound (ii) as follows:

$$
\begin{aligned}
(ii) \quad & \mathbb{E}\left[\left\|(\hat{P} - P)(\hat{P}^\top S\hat{P} + \bar{R})^{-1}P^\top\right\|_2\right] \\
& \leq \left\|\bar{R}^{-1}\right\|_2 \mathbb{E}\left[\left\|\hat{P} - P\right\|_2\right] \|B\|_2 \\
& = \left\|\bar{R}^{-1}\right\|_2 \mathbb{E}\left[\left\|\hat{B} - B\right\|_2\right] \|B\|_2 \\
& \leq \left\|\bar{R}^{-1}\right\|_2 \mathbb{E}\left[\left\|\hat{B} - B\right\|_F\right] \|B\|_2 \\
& \leq \left\|\bar{R}^{-1}\right\|_2 \sqrt{\operatorname{Tr}(K_B)} \|B\|_2 .
\end{aligned}
\tag{82}
$$

We bound (iii) as follows:

$$
\begin{aligned}
(3) \quad & \mathbb{E}\left[\left\|P\left(\left(\hat{P}^\top S\hat{P} + \bar{R}\right)^{-1} - (P^\top SP + \bar{R})^{-1}\right)P^\top\right\|_2\right] \\
& \leq \|P\|_2^2 \, \mathbb{E}\left[\left\|\left(\hat{P}^\top S\hat{P} + \bar{R}\right)^{-1} - (P^\top SP + \bar{R})^{-1}\right\|_2\right] \\
& \leq \|P\|_2^2 \, \mathbb{E}\left[\left\|-(P^\top SP + \bar{R})^{-1}\left(\hat{P}^\top S\hat{P} - P^\top SP\right)\left(\hat{P}^\top S\hat{P} + \bar{R}\right)^{-1}\right\|_2\right] \\
& \leq \|P\|_2^2 \left\|\bar{R}^{-1}\right\|_2^2 \mathbb{E}\left[\left\|\hat{P}^\top S\hat{P} - P^\top SP\right\|_2\right] \\
& = \|P\|_2^2 \left\|\bar{R}^{-1}\right\|_2^2 \mathbb{E}\left[\left\|\hat{P}^\top S(\hat{P} - P) + \left(\hat{P}^\top - P^\top\right)SP)\right\|_2\right] \\
& = \|P\|_2^2 \left\|\bar{R}^{-1}\right\|_2^2 \mathbb{E}\left[\left\|\left(\hat{P}^\top - P^\top\right)S(\hat{P} - P) + P^\top S(\hat{P} - P) + \left(\hat{P}^\top - P^\top\right)SP\right\|_2\right] \\
& \leq \|P\|_2^2 \left\|\bar{R}^{-1}\right\|_2^2 \mathbb{E}\left[\left\|\left(\hat{P}^\top - P^\top\right)S(\hat{P} - P)\right\|_2 + \left\|P^\top S(\hat{P} - P)\right\|_2 + \left\|\left(\hat{P}^\top - P^\top\right)SP\right\|_2\right] \\
& \leq \|P\|_2^2 \left\|\bar{R}^{-1}\right\|_2^2 \left(\|S\|_2 \operatorname{Tr}(K_B) + \|B\|_2 \|S\|_2 \sqrt{\operatorname{Tr}(K_B)} + \|S\|_2 \|B\|_2 \sqrt{\operatorname{Tr}(K_B)}\right) \\
& = \|B\|_2^2 \left\|\bar{R}^{-1}\right\|_2^2 \left(\|S\|_2 \operatorname{Tr}(K_B) + 2\|B\|_2 \|S\|_2 \sqrt{\operatorname{Tr}(K_B)}\right) .
\end{aligned}
\tag{83}
$$

Put everything together, we have the following bound:

$$
\begin{aligned}
\mathbb{E}\left[\left|\hat{J} - J\right|\right] & \leq \|z\|_2^2 \Bigg[ \left\|\bar{R}^{-1}\right\|_2 \left(\operatorname{Tr}(K_B) + \|B\|_2 \sqrt{\operatorname{Tr}(K_B)}\right) \\
& \qquad + \left\|\bar{R}^{-1}\right\|_2 \sqrt{\operatorname{Tr}(K_B)} \|B\|_2 \\
& \qquad + \|B\|_2^2 \left\|\bar{R}^{-1}\right\|_2^2 \left(\|S\|_2 \operatorname{Tr}(K_B) + 2\|B\|_2 \|S\|_2 \sqrt{\operatorname{Tr}(K_B)}\right) \Bigg] \\
& = \|z\|_2^2 \Bigg[ \left(\left\|\bar{R}^{-1}\right\|_2 + \|B\|_2^2 \left\|\bar{R}^{-1}\right\|_2^2 \|S\|_2\right) \operatorname{Tr}(K_B) \\
& \qquad + \left(2\left\|\bar{R}^{-1}\right\|_2 \|B\|_2 + 2\|B\|_2^3 \left\|\bar{R}^{-1}\right\|_2^2 \|S\|_2\right) \sqrt{\operatorname{Tr}(K_B)} \Bigg] \\
& = \|z\|_2^2 \|R^{-1}\|_2 \left(1 + \|B\|_2^2 \|R^{-1}\|_2 \|S\|_2\right) \left(\operatorname{Tr}(K_B) + 2\|B\|_2 \sqrt{\operatorname{Tr}(K_B)}\right) .
\end{aligned}
\tag{84}
$$

$\square$

*Proof of Lagrange multiplier sample complexity result.*

$$U = -R^{-1}B^\top \otimes (I - G_\lambda)^{-1} H_\lambda x_0$$

$$= -\underbrace{\begin{bmatrix} R^{-1}B^\top & & & \\ & \ddots & & \\ & & \ddots & \\ & & & R^{-1}B^\top \end{bmatrix}}_{D_{RB}} (I - G_\lambda)^{-1} H_\lambda x_0$$

$$= -D_{RB}(I - G_\lambda)^{-1} H_\lambda x_0$$

$$= -\underbrace{\begin{bmatrix} R^{-1} & & & \\ & \ddots & & \\ & & \ddots & \\ & & & R^{-1} \end{bmatrix}}_{\bar{R}^{-1}} \underbrace{\begin{bmatrix} B^\top & & & \\ & \ddots & & \\ & & \ddots & \\ & & & B^\top \end{bmatrix}}_{P^\top} (I - G_\lambda)^{-1} H_\lambda x_0$$

$$= -\bar{R}^{-1} P^\top (I - G_\lambda)^{-1} H_\lambda x_0 \tag{85}$$

$$
\begin{aligned}
J &= x_0^\top H^\top \bar{Q} G U + x_0^\top H^\top \bar{Q} H x_0 \\
&= -x_0^\top H^\top \bar{Q} G D_{RB} (I - G_\lambda)^{-1} H_\lambda x_0 + x_0^\top H^\top \bar{Q} H x_0 \\
&= -\underbrace{x_0^\top H^\top \bar{Q} G_d P \bar{R}^{-1} P^\top (I - G_\lambda)^{-1} H_\lambda x_0}_{a} + \underbrace{x_0^\top H^\top \bar{Q} H x_0}_{b} \tag{86}
\end{aligned}
$$

$$\mathbb{E}\left[|\hat{J} - J|\right] = \mathbb{E}\left[\|\hat{J} - J\|_2\right] = \mathbb{E}\left[\|\hat{a} - a\|_2\right] \tag{87}$$

Then

$$
\begin{aligned}
\mathbb{E}\left[\|\hat{a} - a\|_2\right] &= \mathbb{E}\left[\|x_0^\top H^\top \bar{Q} G_d \hat{P} \bar{R}^{-1} \hat{P}^\top (I - \hat{G}_\lambda)^{-1} H_\lambda x_0 - x_0^\top H^\top \bar{Q} G_d P \bar{R}^{-1} P^\top (I - G_\lambda)^{-1} H_\lambda x_0\|_2\right] \\
&= \mathbb{E}\left[\|x_0^\top H^\top \bar{Q} G_d \left(\hat{P} \bar{R}^{-1} \hat{P}^\top (I - \hat{G}_\lambda)^{-1} - P \bar{R}^{-1} P^\top (I - G_\lambda)^{-1}\right) H_\lambda x_0\|_2\right] \\
&\leq \|x_0^\top H^\top \bar{Q} G_d\|_2 \|H_\lambda x_0\|_2 \, \mathbb{E}\left[\|\hat{P} \bar{R}^{-1} \hat{P}^\top (I - \hat{G}_\lambda)^{-1} - P \bar{R}^{-1} P^\top (I - G_\lambda)^{-1}\|_2\right]. \tag{88}
\end{aligned}
$$

Then

$$
\begin{aligned}
&\mathbb{E}\left[\|\hat{P} \bar{R}^{-1} \hat{P}^\top (I - \hat{G}_\lambda)^{-1} - P \bar{R}^{-1} P^\top (I - G_\lambda)^{-1}\|_2\right] \\
&= \mathbb{E}\left[\|(\hat{P} - P) \bar{R}^{-1} \hat{P}^\top (I - \hat{G}_\lambda)^{-1} + P \bar{R}^{-1} \left(\hat{P}^\top (I - \hat{G}_\lambda)^{-1} - P^\top (I - G_\lambda)^{-1}\right)\|_2\right] \\
&= \mathbb{E}\left[\|(\hat{P} - P) \bar{R}^{-1} \hat{P}^\top (I - \hat{G}_\lambda)^{-1} + P \bar{R}^{-1} \left((\hat{P}^\top - P^\top)(I - \hat{G}_\lambda)^{-1} + P^\top (I - \hat{G}_\lambda)^{-1} - P^\top (I - G_\lambda)^{-1}\right)\|_2\right] \\
&= \mathbb{E}\left[\|(\hat{P} - P) \bar{R}^{-1} \hat{P}^\top (I - \hat{G}_\lambda)^{-1} + P \bar{R}^{-1} (\hat{P}^\top - P^\top)(I - \hat{G}_\lambda)^{-1} + P \bar{R}^{-1} P^\top \left((I - \hat{G}_\lambda)^{-1} - (I - G_\lambda)^{-1}\right)\|_2\right] \\
&\leq \underbrace{\mathbb{E}\left[\|(\hat{P} - P) \bar{R}^{-1} \hat{P}^\top (I - \hat{G}_\lambda)^{-1}\|_2\right]}_{(i)} + \underbrace{\mathbb{E}\left[\|P \bar{R}^{-1} (\hat{P}^\top - P^\top)(I - \hat{G}_\lambda)^{-1}\|_2\right]}_{(ii)} \\
&\quad + \underbrace{\mathbb{E}\left[\|P \bar{R}^{-1} P^\top \left((I - \hat{G}_\lambda)^{-1} - (I - G_\lambda)^{-1}\right)\|_2\right]}_{(iii)} \tag{89}
\end{aligned}
$$

Let us rewrite $G_\lambda$ as follows:

$$G_\lambda = L_d + L_u$$

$$= -\underbrace{\begin{bmatrix} QG_0 & 0 & \cdots & 0 \\ QG_1 & QG_0 & \cdots & 0 \\ \vdots & \vdots & \ddots & \vdots \\ QG_{T-1} & QG_{T-2} & \cdots & QG_0 \end{bmatrix}}_{L_{QG}} \underbrace{\begin{bmatrix} BR^{-1}B^\top & & & \\ & \ddots & & \\ & & \ddots & \\ & & & BR^{-1}B^\top \end{bmatrix}}_{D_{BRB}} + L_u$$

$$= -\underbrace{\begin{bmatrix} Q & & & \\ & \ddots & & \\ & & \ddots & \\ & & & Q \end{bmatrix}}_{D_Q} \underbrace{\begin{bmatrix} G_0 & 0 & \cdots & 0 \\ G_1 & G_0 & \cdots & 0 \\ \vdots & \vdots & \ddots & \vdots \\ G_{T-1} & G_{T-2} & \cdots & G_0 \end{bmatrix}}_{L_G} D_{BRB} + L_u$$

$$= -D_Q L_G D_{BRB} + L_u, \tag{90}$$

where

$$L_u = \begin{bmatrix} 0 & A^\top - D(\alpha, 1) & -D(\alpha, 2) & \cdots & -D(\alpha, T-1) \\ 0 & 0 & A^\top - D(\alpha, 1) & \cdots & -D(\alpha, T-2) \\ 0 & 0 & 0 & \cdots & -D(\alpha, T-3) \\ \vdots & \vdots & \vdots & \vdots & \ddots \\ 0 & 0 & \cdots & \cdots & 0 \end{bmatrix} \tag{91}$$

Then using the psd assumption $L_{QG}^{-1}(I - L_u) \succeq 0$,

$$\begin{aligned} \|(I - G_\lambda)^{-1}\|_2 &= \|(I - L_u + D_Q L_G D_{BRB})^{-1}\|_2 \\ &= \| \left[ D_Q L_G \left( (D_Q L_G)^{-1}(I - L_u) + D_{BRB} \right) \right]^{-1} \|_2 \\ &\leq \|(D_Q L_G)^{-1}\|_2 \| \left( (D_Q L_G)^{-1}(I - L_u) + D_{BRB} \right)^{-1} \|_2 \\ &\leq \|(D_Q L_G)^{-1}\|_2 \| \left( (D_Q L_G)^{-1}(I - L_u) \right)^{-1} \|_2 \\ &\leq \|(L_{QG})^{-1}\|_2 \|L_{QG}\|_2 \|(I - L_u)^{-1}\|_2. \end{aligned} \tag{92}$$

We bound $(i)$ as follows:

$$\begin{aligned} &\mathbb{E}\left[ \|(\hat{P} - P)\bar{R}^{-1}\hat{P}^\top(I - \hat{G}_\lambda)^{-1}\|_2 \right] \\ &\leq \mathbb{E}\left[ \|\hat{P} - P\|_2 \|\bar{R}^{-1}\|_2 \|\hat{P}^\top\|_2 \|(I - \hat{G}_\lambda)^{-1}\|_2 \right] \\ &\leq \|\bar{R}^{-1}\|_2 \mathbb{E}\left[ \|\hat{P} - P\|_2 \left( \|\hat{P} - P\|_2 + \|P\|_2 \right) \|(I - \hat{G}_\lambda)^{-1}\|_2 \right] \\ &\leq \|\bar{R}^{-1}\|_2 \mathbb{E}\left[ \|\hat{P} - P\|_2 \left( \|\hat{P} - P\|_2 + \|P\|_2 \right) \right] \|(L_{QG})^{-1}\|_2 \|L_{QG}\|_2 \|(I - L_u)^{-1}\|_2 \\ &\leq \|\bar{R}^{-1}\|_2 \|(L_{QG})^{-1}\|_2 \|L_{QG}\|_2 \|(I - L_u)^{-1}\|_2 \left( \mathbb{E}\left[ \|\hat{B} - B\|_2^2 \right] + \mathbb{E}\left[ \|\hat{B} - B\|_2 \right] \|B\|_2 \right) \\ &\leq \|\bar{R}^{-1}\|_2 \|(L_{QG})^{-1}\|_2 \|L_{QG}\|_2 \|(I - L_u)^{-1}\|_2 \left( \text{Tr}(K_B) + \sqrt{\text{Tr}(K_B)}\|B\|_2 \right). \end{aligned} \tag{93}$$

We bound $(ii)$ as follows:

$$\begin{aligned} &\mathbb{E}\left[ \|P\bar{R}^{-1}(\hat{P}^\top - P^\top)(I - \hat{G}_\lambda)^{-1}\|_2 \right] \\ &\leq \|B\|_2 \|\bar{R}^{-1}\|_2 \|(L_{QG})^{-1}\|_2 \|L_{QG}\|_2 \|(I - L_u)^{-1}\|_2 \sqrt{\text{Tr}(K_B)}. \end{aligned} \tag{94}$$

We Bound $(iii)$ as follows:

$$
\begin{aligned}
(iii) \quad & \mathbb{E}\left[\|P\bar{R}^{-1}P^\top\left((I-\hat{G}_\lambda)^{-1}-(I-G_\lambda)^{-1}\right)\|_2\right] \\
& \leq \|P\|_2^2\|\bar{R}^{-1}\|_2\mathbb{E}\left[\|(I-\hat{G}_\lambda)^{-1}-(I-G_\lambda)^{-1}\|_2\right] \\
& = \|P\|_2^2\|\bar{R}^{-1}\|_2\mathbb{E}\left[\|(I-\hat{G}_\lambda)^{-1}\left(\hat{G}_\lambda-G_\lambda\right)(I-G_\lambda)^{-1}\|_2\right] \\
& \leq \|P\|_2^2\|\bar{R}^{-1}\|_2\|(L_{QG})^{-1}\|_2^2\|L_{QG}\|_2^2\|(I-L_u)^{-1}\|_2^2\mathbb{E}\left[\|\hat{G}_\lambda-G_\lambda\|_2\right] \\
& = \|P\|_2^2\|\bar{R}^{-1}\|_2\|(L_{QG})^{-1}\|_2^2\|L_{QG}\|_2^2\|(I-L_u)^{-1}\|_2^2\mathbb{E}\left[\|L_{QG}(\hat{D}_{BRB}-D_{BRB})\|_2\right] \\
& \leq \|P\|_2^2\|\bar{R}^{-1}\|_2\|(L_{QG})^{-1}\|_2^2\|L_{QG}\|_2^3\|(I-L_u)^{-1}\|_2^2\mathbb{E}\left[\|\hat{B}R^{-1}\hat{B}^\top-BR^{-1}B^\top\|_2\right] \\
& = \|P\|_2^2\|\bar{R}^{-1}\|_2\|(L_{QG})^{-1}\|_2^2\|L_{QG}\|_2^3\|(I-L_u)^{-1}\|_2^2\mathbb{E}\left[\|(\hat{B}-B)R^{-1}(\hat{B}-B)^\top\right. \\
& \quad\quad \left.+(\hat{B}-B)R^{-1}B^\top+BR^{-1}(\hat{B}-B)^\top\|_2\right] \\
& \leq \|B\|_2^2\|\bar{R}^{-1}\|_2^2\|(L_{QG})^{-1}\|_2^2\|L_{QG}\|_2^3\|(I-L_u)^{-1}\|_2^2\left(\mathrm{Tr}(K_B)+2\|B\|_2\sqrt{\mathrm{Tr}(K_B)}\right).
\end{aligned}
\tag{95}
$$

Using $\|(I-L_u)^{-1}\|_2=1$ and putting everything together we have

$$
\begin{aligned}
& \mathbb{E}\left[|\hat{J}-J|\right] \\
& \leq \|z\|_2\|H_\lambda x_0\|_2\|R^{-1}\|_2\|\mathbb{L}\|_2\left(1+\|B\|_2^2\|R^{-1}\|_2\|\mathbb{L}\|_2\|L_{QG}\|_2\right)\left[\mathrm{Tr}(K_B)+2\|B\|_2\sqrt{\mathrm{Tr}(K_B)}\right], \\
& where \; \|\mathbb{L}\|_2=\|(L_{QG})^{-1}\|_2\|L_{QG}\|_2.
\end{aligned}
\tag{96}
$$

$\square$

## D.5. Proof of Corollary 3.4

*Proof.* We first expand $K_B$ as follows

$$
\begin{aligned}
K_B & = \frac{1}{N}\left(\phi^\top\phi\right)^{-1}\phi^\top\sigma_w^2 I\phi\left(\phi^\top\phi\right)^{-1} \\
& = \frac{\sigma_w^2}{N}\left(\phi^\top\phi\right)^{-1} \\
& = \frac{\sigma_w^2}{N}\left[\sum_{i=1}^p\phi_i^\top\phi_i\right]^{-1},
\end{aligned}
\tag{97}
$$

where

$$
\sum_{i=1}^p\phi_i^\top\phi_i = \begin{bmatrix} \sum_{i=1}^p u_0^i u_0^{i\top} & \cdots & 0 \\ \vdots & \ddots & \vdots \\ 0 & \cdots & \sum_{i=1}^p u_0^i u_0^{i\top} \end{bmatrix}.
\tag{98}
$$

Then we have

$$\text{Trace}(K_B) = \frac{\sigma_w^2}{N} \mathbb{E} \left[ \text{Trace} \left( \left[ \sum_{i=1}^{p} \phi_i^\top \phi_i \right]^{-1} \right) \right]$$

$$= \frac{n\sigma_w^2}{N} \mathbb{E} \left[ \text{Trace} \left( \left( \sum_{i=1}^{p} u_0^i u_0^{i\top} \right)^{-1} \right) \right]$$

$$= \frac{n\sigma_w^2}{N} \text{Trace} \left( \mathbb{E} \left[ \left( \sum_{i=1}^{p} u_0^i u_0^{i\top} \right)^{-1} \right] \right). \tag{99}$$

$$(1) \begin{cases} u_0^i \sim \mathcal{N}(0, I) \implies \sum_{i=1}^{p} u_0^i u_0^{i\top} \sim \mathcal{W}(I, p), \text{ Wishart Distribution (Wishart, 1928)} \\[2mm] \implies \left( \sum_{i=1}^{p} u_0^i u_0^{i\top} \right)^{-1} \sim \mathcal{W}^{-1}(I, p), \text{ Inverse Wishart Distribution} \\[2mm] \implies \mathbb{E} \left[ \left( \sum_{i=1}^{p} u_0^i u_0^{i\top} \right)^{-1} \right] = \frac{I}{p - m - 1}, \quad \text{for } p > m + 1. \end{cases}$$

$$\text{Trace}(K_B) \overset{(1)}{=} \frac{n\sigma_w^2}{N} \text{Trace} \left( \frac{I}{p - m - 1} \right)$$

$$= \frac{nm\sigma_w^2}{N(p - m - 1)}, \quad \text{for } p > m + 1. \tag{100}$$

$\square$

# E. Experiment Details

## E.1. Model Architecture

Our proposed Fractional-Order Learning for Optimal Control Framework (FOLOC) is a deep learning framework designed for fractional-order optimal control tasks, combining the strengths of nerual operater, sequential modeling, and parameter regression. The architecture is structured into four key components, enabling end-to-end learning of complex control dynamics from system states and parameters. Fig. 1 illustrates the overall pipeline.

### E.1.1. System Identification

The system identification module is designed to infer fractional-order time-invariant system parameters, including the fractional-order $\alpha$ of size $n$ and the constant matrices $A$ and $B$ of sizes $n \times n$ and $n \times m$ respectively, as introduced in Eq. 6. These parameters are directly inferred from the observed control input space $\mathcal{X}$, which is formed by concatenating $x_k$ and $u_k$, where $x_k \in \mathbb{R}^n$ is the state vector and $u_k \in \mathbb{R}^m$ is the system input. This component leverages a hybrid architecture to accommodate both static and temporally correlated systems. Specifically, we use an MLP with residual skip connections and a Sequence Model (e.g., Recurrent Neural Networks, Gated Recurrent Unit Networks, or Long Short-Term Memory Networks) as the backbone of system identification module.

For non-Markovian systems with long-range temporal temporal dependencies, sequential encoders—such as bidirectional LSTMs, GRUs, or RNNs—are adopted. These models process the input sequence autoregressively, capturing latent temporal patterns through hidden state evolution. Bidirectional processing further enables context aggregation from both past and future observations within the horizon T. The regressor outputs a flattened tensor, which is decomposed into $A$, $B$ and $\alpha$ via learned linear projections, ensuring dimensional consistency with the underlying physical system. We formalize this module as follows

For non-Markovian systems, bidirectional LSTMs model temporal dependencies:

$$h_t, c_t = \text{LSTM}\big([x_t; u_t], (h_{t-1}, c_{t-1})\big), \tag{101}$$

where $h_t \in \mathbb{R}^{2d}$ (bidirectional hidden states). The final state $h_T$ aggregates sequence information.

Then we apply a stack of residual MLP blocks processes final state $h_T$:

$$\text{h}^{(0)} = h_T, \text{h}^{(l)} = \text{BatchNorm}\Big(\sigma\big(W^{(l)} \text{h}^{(l-1)} + b^{(l)}\big) + \text{h}^{(l-1)}\Big), \tag{102}$$

where $\sigma$ is an activation function (e.g., ReLU) and $1, ..., L$ indexes the residual blocks.

The regressor outputs a flattened tensor $\theta \in \mathbb{R}^{n^2+nm+n}$, which can be decomposed into $A$, $B$ and $\alpha$

$$\theta = W_{out}\text{h}^{(\text{L})}, \tag{103}$$
$$A = Reshape(\theta_{1:n^2}), B = Reshape(\theta_{n^2+1:n^2+nm}), \alpha = \theta_{n^2+nm+1:n^2+nm+n} \tag{104}$$

The module minimizes a multi-task regression loss:

$$\mathcal{L}_{\text{s}} = \|A - A^*\|_2^2 + \|B - B^*\|_2^2 + \|\alpha - \alpha^*\|_2^2. \tag{105}$$

This module is trained under a multi-task objective, jointly optimizing the reconstruction of system parameters while preserving their interpretable structure (e.g., enforcing A as a state transition matrix). By unifying static and sequential modeling paradigms, the regressor adapts to diverse dynamical regimes, laying a foundation for downstream control synthesis.

### E.1.2. TEMPORAL EMBEDDING FOR $A$ MATRIX EVOLUTION

In order to obtain the temporal embedding of matrix $A$, we propose an embedding estimation module that transforms the estimated fractional-order system parameters into time-aware representations. These representations capture both the spectral properties of the dynamics and their temporal evolution. By addressing the challenge of modeling parameter drift in partially observed systems, this approach maintains physical consistency throughout.

Our architecture implements a novel parameterization of the system matrix $A$, explicitly encoding hierarchical damping effects. This allows for both stability guarantees and adaptive temporal evolution. Building on Eq. 15, the module constructs time-dependent embeddings $A_t^{emb} \in \mathbb{R}^d$ through a dual-branch architecture.

Let $A \in \mathbb{R}^{n \times n}$ and $\alpha \in \mathbb{R}^n$ denote the estimated constant matrix and fractional-order from the system identification module. To compute the embedding of the first token $A_0$, we decompose the system matrix $A_\alpha = A - \text{diag}(a)$ through spectral analysis:

$$A_\alpha = V \Lambda V^{-1}, \quad \Lambda = \text{diag}(\lambda_1, \ldots, \lambda_n), \tag{106}$$
$$h_{spec} = \sigma\big(W_\gamma\big[\text{Re}(\Lambda), \text{Im}(\Lambda)\big]\big), \tag{107}$$

where $\lambda_i \in \mathbb{C}$ are eigenvalues encoding system stability. A residual MLP processes the concatenated real/imaginary components $[\text{Re}(\Lambda), \text{Im}(\Lambda)]$ to produce base embeddings $A_0$.

In parallel, time embeddings $e_t$ are generated via an embedding layer that encodes sequential time indices. These are concatenated with embeddings derived from $\alpha$, enabling the model to capture both temporal and parametric variations:

$$h_{\text{state}} = \phi_x(e_t) \oplus \phi_\alpha(\alpha), \quad \phi_x, \phi_\alpha : \text{Linear Projections.} \tag{108}$$

Next, the adaptive fused embeddings pass through a series of stacked residual multi-layer perceptron (MLP) blocks with normalization and an activation function (e.g., ReLU or GELU). This produces the final embeddings that serve as inputs to the temporal processor. This residual architecture enhances the model's capacity to learn complex representations while preserving information from earlier layers:

$$A^{emb} = \text{ResMLP}\big([h_{spec} \oplus h_{\text{state}}]; W_l, b_{l\,l=1}^L\big), A^{emb} \in \mathbb{R}^{T \times hidden\_size} \tag{109}$$

This embedding mechanism enables the model to generate time-dependent parameter representations without directly relying on the input state vector $x$. By encoding dynamic properties through eigenvalues and temporal embeddings, the model effectively captures system behavior over time, making it suitable for tasks such as time-series forecasting and control parameter prediction.

### E.1.3. SEQUENCE ENCODER PROCESSOR

The Sequence Processor translates spectral-temporal embeddings $A_t^{emb}$ into latent states $G_t^{enc}$ that encode the system's temporal evolution, enabling adaptive control under time-varying dynamics. The sequence processor computes latent states $G_t^{enc}$ through a recursive hierarchical structure, enabling efficient modeling of temporal dependencies while preserving stability, rooted in the governing Eq. 14.

we implements this recurrence through a deep encoder architecture, which projects the raw recursive computation into a latent space. For systems requiring global temporal context, we use Transformer as backbone:

$$G_k^{enc} = TransformerEncoder(A_k^{emb} + Pk), \tag{110}$$

where $P_k$ is a positional encoding encoding the step $k$. Multi-head self-attention layers implicitly approximate the summation in Eq. 14 by learning attention weights that mirror the hierarchy $A_j$.

### E.1.4. STACK MLPs FOR FOURIER NEURAL OPERATOR

The final stage of our architecture synthesizes hierarchical system representations into spatiotemporal control signals through a two-step process: feature unification via the Stack MLPs and spectral control synthesis via the Fourier Neural Operator (FNO). This combination enables efficient learning of distributed control laws while preserving physical consistency. Given temporal states $G_k^{enc}$, system parameters $B$, LQR cost matrices $Q,R$, and residual skip connection input $[x; u]$, the transformation constructs an enriched input tensor $\mathcal{X} \in \mathbb{R}^{T \times d}$ for the FNO:

$$\chi_t = \phi_G(G_t) \oplus \phi_B(B) \oplus \phi_Q(Q) \oplus \phi_R(R) \oplus x_t \oplus u_t \tag{111}$$

where $\phi$ are a series of stacked residual multi-layer perceptron (MLP) blocks and $\oplus$ denotes channel-wise concatenation. Next, The FNO processes $\mathcal{X}$ through spectral convolutions and residual connections as we introduced in Section 3.3:

$$\mathcal{F}(\chi)(k) = \mathbf{W}(k) \cdot \chi(k) + \mathbf{b}(k), \quad k \in \mathbb{Z}^d \tag{112}$$

where $\chi(k) = \mathcal{F}(\chi)$ is the Fourier transform of the input, and $\mathbf{W}, \mathbf{b}$ are learnable parameters in frequency space. Only low-frequency modes $\|\mathbf{k}\| \leq k_{\max}$ are retained, enforcing spectral sparsity.

Then, we apply the inverse transformation to predict the optimal control $U$:

$$U = \mathcal{F}^{-1}\left(\prod_{l=1}^{L}\left((W_l \mathcal{F} + K_l)\mathcal{F}(\chi)\right)\right) + \text{MLP}(\chi), \tag{113}$$

where $K_l$ are local kernel integrations in physical space.

The module minimizes the deviations between predicted $U_t$ and optimal control trajectories $U_t^*$:

$$\mathcal{L}_u = \frac{1}{Tm}\sum_{t=1}^{T}\|U_t - U_t^*\|_2^2. \tag{114}$$

### E.1.5. COMPOSITE LOSS FORMULATION

The training objective combines fractional-order system parameter regression and control prediction errors through a multi-task loss function, enabling joint optimization of system identification and control synthesis:

$$\mathcal{L}_{\text{total}} = \underbrace{\mathcal{L}_s}_{\text{System Identification}} + \underbrace{\mathcal{L}_u}_{\text{Control Prediction}}. \tag{115}$$

This unified framework successfully marries physics-guided feature engineering with data-driven spectral learning, enabling real-time optimal control in high-dimensional dynamical systems and the unified loss formulation enables end-to-end learning of control policies while maintaining physically consistent parameter estimates—a critical requirement for deployment in safety-critical dynamical systems.

remark: Specifically, the system parameters $A, B$ and $\alpha$ can be extracted from an intermediate layer in the network because of the underlying loss function. The intermediate variable $A_k$ is approximated using a shallow neural network $\mathcal{P}$ such that $A_k = \mathcal{P}(A, \alpha, t)$ and the variable $G_k$ is reduced to $\mathbb{R}^n$ instead of $\mathbb{R}^{n \times n}$. Similarly, the cost matrices $Q$ and $R$ are transformed into reduced representations via local transformation operators $\mathcal{N}_Q : \mathbb{R}^{n \times n} \to \mathbb{R}^{d_Q}$ and $\mathcal{N}_R : \mathbb{R}^{n \times m} \to \mathbb{R}^{d_R}$. The block Toeplitz structure of $G_\lambda$ indicates that it can be fully represented using the parameters $Q, R, A, B, \alpha, x_0$ and $\{G_k\}_{k=0}^T$. Using this property, the input dimension of the FNO is determined as $d_a = T \times (2n + m + d_Q + d_R + d_B)$, where the extra terms $n$ and $m$ correspond to the trajectory and control input dimensions, respectively. These additional features empirically enhance the performance of the FNO by providing more comprehensive representations of the underlying dynamics.

## E.2. Synthetic Data

The synthetic data is generated as follows:
State Transition Matrix ($A$): Rondomly sampled from a uniform distribution and normalized to ensure stability by scaling with its spectral radius.
Control Input Matrix ($B$): Randomly sampled from a uniform distribution.
Initial State ($x_0$): Randomly sampled from a standard normal distribution.
Control Inputs ($u$): Sequence of uniformly random inputs.
Cost Matrices ($Q, R$): Symmetric positive semi-definite matrices constructed via $M^\top M$ for some uniformly random $M$.

## E.3. Real-World System Dynamics

In our experiments, we simulate the system dynamics to generate trajectories and optimal control sequences by randomly choosing an initial state and control inputs and then using model predictive control (MPC) to find the optimal control sequence given the quadratic cost matrices $Q$ and $R$.

### E.3.1. CART POLE SYSTEM DYNAMICS

The dynamics of the cart pole is as follows:

$$\ddot{x} = \frac{m_p l \left( \dot{\theta}^2 \sin\theta - \ddot{\theta} \cos\theta \right) + F}{m_c + m_p},$$

$$\ddot{\theta} = \frac{g \sin\theta + \cos\theta \left( \frac{-m_p l \dot{\theta}^2 \sin\theta - F}{m_c + m_p} \right)}{l \left( \frac{4}{3} - \frac{m_p \cos^2\theta}{m_c + m_p} \right)}, \tag{116}$$

where $\ddot{x}$ is the horizontal acceleration of the cart, $\ddot{\theta}$ is the acceleration of the pole, $m_c$ is the mass of the cart, $m_p$ is the mass of the pole, and $g$ is the acceleration due to gravity. The control input is the external force $F$ applied to the center of mass of the cart.

### E.3.2. QUADROTOR SYSTEM DYNAMICS

The quadrotor dynamics is as follows:

$$\dot{p} = v,$$

$$\dot{q} = \frac{1}{2} q \otimes \begin{bmatrix} 0 \\ \omega \end{bmatrix},$$

$$m\dot{v} = mg + f + f_a,$$

$$\mathcal{J}\dot{\omega} = -\omega \times \mathcal{J}\omega + \tau. \tag{117}$$

Here, $p \in \mathbb{R}^3$ and $v \in \mathbb{R}^3$ represent the position and velocity in the inertial frame, $\omega \in \mathbb{R}^3$ is the angular velocity, and the quaternion $q$ describes the orientation with $\otimes$ denoting quaternion multiplication. The mass of the quadrotor is given by $m$, and $\mathcal{J}$ denotes the inertia matrix. Gravity is denoted by $g$, the aerodynamic force is denoted by $f_a \in \mathbb{R}^3$, the total thrust and body torques are denoted by $f \in \mathbb{R}^3$ and $\tau \in \mathbb{R}^3$.

## F. Training Parameters

The experiments utilizing GPU acceleration on an NVIDIA A100 80GB PCIe GPU. The experiments are conducted on a machine running Ubuntu 22.04.5 LTS with an Intel(R) Xeon(R) Platinum 8358 CPU (2.60 GHz), featuring 128 cores and support for 256 concurrent threads.

The training process is configured to optimize model performance using the Adam optimizer with a learning rate of $10^{-3}$. The model is trained for 300 epochs with a batch size of 128, Learning rate scheduling is managed by a `ReduceLROnPlateau` scheduler, which monitors validation loss and reduces the learning rate by a factor of 0.1 after 5 epochs without improvement. The scheduler uses a relative threshold of 0.0001 to detect improvement. The minimum learning rate is set to zero, and early learning stabilization is facilitated by an epsilon value of $1 \times 10^{-8}$. The loss function based on the $L_p$-norm.

To ensure robust training, data normalization is applied, with both the input data and control signals ($U$) included in the modeling process. The architecture supports both encoder-only and sequence-to-sequence configurations; however, in this setup, the sequence-to-sequence mode is disabled.

## G. Metrics

**Mean Squared Error (MSE).** The Mean Squared Error is given by

$$\mathcal{L}_{\text{MSE}}(\hat{y}, y) = \frac{1}{n} \sum_{i=1}^{n} (y_i - \hat{y}_i)^2, \tag{118}$$

where $n$ denotes the number of samples, $y_i \in \mathbb{R}$ is the ground truth value, and $\hat{y}_i \in \mathbb{R}$ is the predicted value for the $i$-th sample.

**Mean Absolute Error (MAE).** The Mean Absolute Error is given by

$$\mathcal{L}_{\text{MAE}}(\hat{y}, y) = \frac{1}{n} \sum_{i=1}^{n} |y_i - \hat{y}_i|, \tag{119}$$

where $n$, $y_i$, and $\hat{y}_i$ are defined as in the MSE formulation.

## H. Ablation Study

In this section, we provide further analyses to uncover how the choice of encoder models and weight of identification loss affect the performance of the FOLOC framework.

### H.1. Varying Sequence Encoder Module

To investigate the performance of different encoder layers on fractional-order system, we choose four sequence encoder variants—Transformer, RNN, LSTM, and GRU. We also test FOLOC framework by replacing the sequence encoder module with a MLP. Our analysis of sequence encoder architectures reveals statistically significant differences in performance and stability across models. The results are given in Table 7. The Transformer encoder demonstrates superior performance, achieving the lowest mean squared error (MSE = 0.00495 ± 0.00010) with minimal variance, underscoring its robustness to initialization. These results highlight the ability of our model's self-attention mechanisms to handle tasks involving global temporal dependencies. Based on our experiments, the Transformer architecture mitigates the risk of gradient vanishing or explosion by distributing information through attention, effectively stabilizing the optimization process and demonstrating strong suitability for safety-critical control tasks.

*Table 7.* Ablation study of different sequence encoders.

| Model | MSE ($10^{-3}$) | MAE ($10^{-2}$) |
|---|---|---|
| **Transformer** | **4.95 $\pm$ 0.10** | **1.05 $\pm$ 0.10** |
| RNN | 6.00 $\pm$ 1.46 | 1.15 $\pm$ 0.17 |
| LSTM | 5.54 $\pm$ 1.49 | 1.12 $\pm$ 0.18 |
| GRU | 6.78 $\pm$ 2.36 | 1.29 $\pm$ 0.34 |
| MLP | 5.91 $\pm$ 1.57 | 1.20 $\pm$ 0.21 |

*Table 8.* Varying identification weight. All reported values are in units of $10^{-3}$.

| Metrics | $\lambda_w = 0$ | $\lambda_w = 0.1$ | $\boldsymbol{\lambda_w = 0.2}$ | $\lambda_w = 0.3$ | $\lambda_w = 0.4$ | $\lambda_w = 0.5$ |
|---|---|---|---|---|---|---|
| MSE | 7.54$\pm$3.11 | 4.95$\pm$0.10 | **4.89$\pm$0.14** | 5.19$\pm$0.65 | 5.84$\pm$1.36 | 5.08$\pm$0.29 |
| MAE | 14.1$\pm$0.39 | 10.5$\pm$0.10 | **10.3$\pm$0.07** | 11.0$\pm$0.83 | 11.4$\pm$1.69 | 11.1$\pm$0.54 |

## H.2. Varying Identification Weight

We evaluate the performance of the FOLOC framework by varying the system identification weight $\lambda_w$ over the values $\{0, 0.1, 0.2, 0.3, 0.4, 0.5\}$. Experimental results show that FOLOC's performance remains approximately the same, with a mean MSE of 0.56% (standard deviation: 0.10%) and a mean MAE of 1.14% (standard deviation: 0.14%). The results are given in Table 8.

## I. Baseline

To the best of our knowledge, no existing end-to-end deep learning framework addresses the optimal control problem under fractional-order dynamical systems, nor is there a theoretical framework that integrates end-to-end learning with FOLTI system identification and optimal control laws. Due to the principle that fractional-order calculus generalizes integer-order calculus, we did some experiments to compare our proposed methods with the traditional theoretical end-to-end approach for optimal control in LTI systems. The end-to-end learning framework for LTI systems involves system identification using ordinary least squares, followed by solving the optimal control problem through dynamic programming (Anderson & Moore, 2007).

## J. Additional Results

### J.1. Test Loss of Different Time Horizons

We present the MSE test loss across different time horizons as shown in Fig. 5, reveals critical insights into its temporal generalization capabilities. With performance steadily improving as the horizon extends, it reflects the FOLOC framework capacity to suppress high-frequency errors through spectral filtering while gradually resolving low-frequency dynamics. $T = 128$ not only shows a smooth reduction in MSE but also achieves the lowest MSE loss among all tested horizons, indicating that the model better captures long-range temporal dependencies when provided with more context. The finding that extending the prediction window enhances both model robustness and accuracy confirms that, when given sufficient historical information, our approach is better equipped to handle complex dynamics of fractional-order systems.

### J.2. Results of Different Fractional-Order Systems

To evaluate the model's robustness across diverse fractional-order systems, we tested FOLOC framework on a dataset constructed from 10,000 unique systems with varying parameters. Our approach demonstrates consistent performance on this heterogeneous evaluation set, achieving a mean squared error (MSE) of $(8.0025 \pm 1.1420) \times 10^{-3}$ and mean absolute error (MAE) of $(9.4546 \pm 0.7698) \times 10^{-3}$, indicating stable generalization despite substantial system diversity. These results reflect the architecture's capacity to adapt to different fractional-order systems without task-specific fine-tuning. Such findings align with theoretical expectations, as the spectral-temporal embeddings and the FOLOC framework inherently decouple system-specific features from control synthesis, enabling robust out-of-distribution operation.

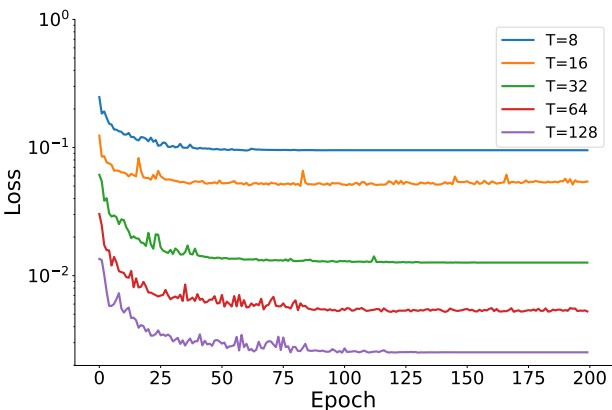

*Figure 5.* Test MSE loss under each epochs.

### J.3. Comparison of Inference Time

The experimental results demonstrate that FOLOC framework achieves superior computational efficiency compares to LTI model. Specifically, FOLOC framework requires only 1.215 ms per sample for inference, which represents a significant $44.6\%$ reduction in processing time compared to LTI model's 2.1935 ms per sample.

*Table 9.* Model inference time comparison.

| Model | Inference time (ms/sample) |
|---|---|
| LTI Model | 2.1935 |
| FOLOC | 1.215 |

