# OpenReview forum: "End-to-End Learning Framework for Solving Non-Markovian Optimal Control"
_ICML.cc/2025/Conference — ICML 2025 poster_

### Official Review · Reviewer_2GMe · 2025-03-04

**Overall Recommendation:** 3

**Summary:**

This paper devises a learning-based approach to solve linear quadratic regulator (LQR) problems for fractional-order linear time-invariant systems. Here, the fractional order means the system state transition is not entirely dependent on the current state but also depends on previous states (Eq(7) in the paper). This approach involves two procedures: system identification and controller synthesis. In system-ID, given a set of traces from the plant runs, the key coefficients for the system dynamics (A, B matrices) and the fractional factor ($\alpha$) are estimated. In the control synthesis step, given the A, B, $\alpha$, and LQR spec ($Q$, $R$ matrix), the control sequence can be derived using Fourier Neural Operators (FNO). A theorem regarding sample complexity is given with proof in the paper. The authors conduct experiments in both synthetic data and two nonlinear control problems to demonstrate that the proposed method is scalable, robust, and computation-efficient.

## update after rebuttal
I have read the author's response as well as other reviewers' comments. Overall, I think the work is novel and is okay to be accepted by ICML, though a few flaws exist. I updated my final ratings as "weak accept."

**Claims And Evidence:**

Yes.

**Essential References Not Discussed:**

Not that I am aware of.

**Experimental Designs Or Analyses:**

Yes. I checked all of them.

**Methods And Evaluation Criteria:**

Yes.

**Other Comments Or Suggestions:**

For Figure 2, you might want to have a recap for the complexity you mentioned - and also in Theorem 3.3, you mentioned two complexities (one for least-square fitting; the other one for the Lagrange-multiplier solution). Which one do you refer to here?

For Figure 4, you don't need two separate y-axis, since they are both errors, and their scale ranges are roughly the same (also I notice that they are actually not aligned well - if you compare the positions of "0.00" for the two measures, you will find out they actually not on the same horizontal level - similar things happens for the "0.10" value - this is not rigorous when plotting something like this).

The notation is not very consistent: in figure 1, you use $\hat{U}^o$ for network output and $U^o$ for precomputed one, but in Eq(29) you use $U$ for network output and $\hat{U}_\theta$ for the pre-computed one.

**Other Strengths And Weaknesses:**

Pros:
1. The proposed method is novel, and I see a good combination of ML with the control theory domain.
2. Theoretical guarantees are provided.

Cons:
1. Clarity: The paper is not very easy to understand. I have put some of my comments/questions in the sections below.
2. Lack baselines: I don't see a clear advantage of the proposed approach - why should we use this method instead of some other existing methods? There are many ways to stabilize a system, especially for systems used in this paper, like the cart pole and quadrotor. I suggest the authors to compare to some of the representatives (I could name a few, model-free/model-based RL, CEM/CMA-ES, MPC, PID, iLQR, Hamilton–Jacobi–Bellman equations; but you are encouraged to try more).
3. OOD: This approach requires sys-id from trajectories, so the training trajectory distribution should support well around the LQR-trajectory distribution - which might not be possible if you don't start with a good MPC to roll out these demonstration trajectories. But if you can have a good MPC to drive the system to track on target, why bother to use this learning-based LQR? This fact limits the application to a more general setup.

**Questions For Authors:**

I am still confused about the general pipeline: so at the beginning, you only have a bunch of trajectories. Then you estimate the A, B, $\alpha$ using least-square, and then you can compute the optimal control sequence $U^{\theta}$ using Eq(16) or Eq(17). Based on your loss in Eq(29), you need to have these (the A,B,$\alpha$ and $U$) ready before learning the NN. Then why do you still need to learn anything using the NN?

Line 212-213: "where $p$ and $N$ denote the numbers of samples..." What do you mean here? Are they both denoting for the number of samples?

"...and assuming the diagonal elements of the system matrix A are known..." If you are doing sys-id, why can we assume this in real-world applications? Can you comment on how strong/weak this assumption is?

Why not train the two stages (sys-id; control-synthesis) separately? I don't see they are coupled in any way, so you should be able to first train sys-id and then train the controller. Using $\lambda$ here in Eq(28) will trade off both performances.

I don't understand in Table1 the last row. Why will increasing the sample size decrease the performance? Can you explain this?

Why do you need a separate SEQ model to handle tokens since you already have an RNN structure to handle temporal data? I don't understand the design choice of blending both RNN and Transformer-based sequence encoder. Do you have ablation studies to support this design choice? (I saw in appendix-Table4 you compare different SEQ model archs I guess, but I don't the result to support your usage of RNN in the pipeline).

**Relation To Broader Scientific Literature:**

This paper proposes a novel learning-based approach to control dynamical systems with long-term time dependencies, which can potentially solve non-Markovian problems. The experiments in this paper go beyond simple linear dynamics and consider high-dimension nonlinear systems, which shows potential adaptation to a broader range of applications.

**Theoretical Claims:**

Yes. I checked all of them.

---

> ### Author Rebuttal · Authors · 2025-04-01
>
> Thank you for your careful reading and constructive suggestions. Below, we provide a detailed response regarding your concerns and questions.
>
> **Baselines**: As detailed in our response to **Reviewer cBiX**, we have conducted extensive comparisons between FOLOC and several widely used methods, including MBRL, CMA-ES, MPC, PID, and iLQR.
>
> **Why learning based LQR?** (i) FOLOC generalizes across different FOLTI systems (Appendix J.2), predicting system parameters and optimal controls (OCs) directly from trajectories. **System parameters are not used during testing on synthetic data, and are not used entirely during both training and testing on real-world dynamics**. (ii) It enables fast and accurate real-time control. Although closed-form solutions (Eqs. 16 and 17) exist given full system knowledge, they involve large-scale matrix inversions and become impractical for long horizons or high-dimensional systems. MPC, though an alternative, also struggles with the memory effects in fractional-order systems. FOLOC overcomes these challenges by learning a direct mapping from trajectories and cost matrices to OCs.
>
> **Figures**: We modified our manuscript to address your concerns about figures: we (i) clarified in Fig. 2’s caption that the sample complexity refers to the Lagrange multiplier-based solution, (ii) updated Fig. 4 by removing the secondary y-axis and aligning both error metrics on a shared axis, and (iii) revised Fig. 1’s notation in alignment with Eq. (29).
>
> **Joint optimization pipeline**: The sys-id loss is a physical constraint for our model that encourages the model to accurately identify the system, further ensuring accurate control, the ultimate goal of our method. During the training period, we use the sys-id label as guidance to ensure accurate sys-id, and **we do not need these labels in the test period**. Some methods, such as PINN [1], couple multiple loss terms by incorporating physical constraint losses. This helps the system learn representations that are both physically grounded and task-relevant, reducing error propagation and avoiding reliance on purely minimizing prediction error. For real-world datasets where the system parameters are unknown, we can first pretrain the model on synthetic datasets with known parameters(**$A$, $B$, $\alpha$**). Then, we do not need sys-id labels(set sys-id loss weight $λ=0$) and finetune the model on the real-world dataset. We'll provide results for $λ=0$ for comparison in Appendix H.1.
> |Metrics|MSE(×10⁻³)|MAE(×10⁻²)|
> |-|-|-|
> |$λ=0$|7.54±3.11|1.41±0.39|
>
> **Notations $p$ and $N$**: We have clarified in the revised version that $p$ denotes the number of initial conditions, and $N$ denotes the number of trials adding Gaussian noise to fractional-order dynamics from these $p$ initial conditions.
>
> **Assumption**: This is a strong assumption and represents a limitation of the theoretical sys-id approach. It serves a similar role to assumptions commonly made in classical sys-id, where the exact (integer) order of the system is often presumed known in advance. In the context of fractional-order systems, estimating the system order—i.e., the fractional derivatives—is substantially more challenging, and to the best of our knowledge, no existing method provides theoretical guarantees in a general setting. To address this limitation, we propose FOLOC framework, which can give accurate OC solutions without this assumption.
>
> **Performance decrease**: The decrease of the performance may be caused by the **Joint training loss trade-off** and **Stochastic training dynamics and over-regularization**, Despite this the variation in MSE/MAE across training sizes is small (e.g., 5.18 → 5.22 × 10⁻³ in MSE) indicate that the model exhibits both robustness and stability across different settings even with a few sample data
> The results are as follows, which show a significant drop in performance using fewer training samples.(i.e., 500) (All reported values are in units of 10⁻³.)
> |Metrics|MSE|MAE|
> |-|-|-|
> |$N_{T}$ = 500|10.5±3.17|22.8±1.35|
>
> **Designing of model architecture**: Align with theoretical derivation in Eqs. (13)–(15), The system parameters($A, B, \alpha$) in physical equations of the fractional-order system reveal a recursive relationship within the sequence. We want to use RNN-based to capture the recursive dependency across time steps. In Eq. (13), the sequence $A_j$ is derived from the system parameters A and $\alpha$, and we employ a sequential model to estimate this computation. We also experimented with replacing the RNN with MLP to predict sys-id parameters. The performance degradation highlights the necessity of using RNN-based models.  (All reported values are in units of 10⁻³.)
> |Metrics|MSE|MAE|
> |-|-|-|
> |RNN + MLP|4.95±0.09|10.5±0.10|
> |MLP Only|5.91±1.57|12.0±2.12|
>
> [1] Raissi, et al. "Physics-informed neural networks: A deep learning framework for solving forward and inverse problems involving nonlinear partial differential equations."

---

### Official Review · Reviewer_cBiX · 2025-03-16

**Overall Recommendation:** 3

**Summary:**

In summary, the paper contributes a novel end-to-end, data-driven framework for optimal control of non-Markovian (fractional-order) systems by uniting advanced control theory with deep learning techniques, backed by both theoretical guarantees and empirical performance.

**Claims And Evidence:**

The paper provides a strong combination of rigorous theoretical derivations, algorithmic formulations, and empirical experiments that support many of its key claims. In particular, the extension of LQR theory to fractional-order systems is well-founded, and the proposed FOLOC framework is validated on both synthetic and real-world benchmarks. The sample complexity analysis also provides theoretical backing to the learning guarantees.

While the experiments demonstrate robustness and efficiency on moderate-dimensional systems, further evidence would be valuable to confirm that the framework scales effectively to very high-dimensional settings. Additional experiments or comparisons in such regimes would strengthen this claim. The paper asserts that the end-to-end framework enables computational efficiency and fast, real-time control evaluation. Although the empirical results indicate computational advantages, more detailed benchmarks or runtime comparisons with existing methods would provide clearer support for this claim.

**Essential References Not Discussed:**

As said before, I am not familiar with the literature of the FOLTI systems, so I cannot provide anything specific about possible missing references. However, I find that the authors provide a great literature review in the supplemental material.

**Experimental Designs Or Analyses:**

I think the experimental designs are nice but it may strengthen the paper if the authors can provide comparisons to some existing methods.

**Methods And Evaluation Criteria:**

I think the proposed method makes sense for the scope of problems considered in the paper, i.e. solving for optimal control for fractional-order dynamical systems.

**Other Comments Or Suggestions:**

I am not sure why the header of the paper is "Submission and Formatting Instructions for ICML 2025". Maybe it's a formatting misprint and I hope the authors will correct it in the revised draft.

**Other Strengths And Weaknesses:**

N/A

**Questions For Authors:**

I do not have any questions for the authors.

**Relation To Broader Scientific Literature:**

I do not know much about the FOLTI systems and the methods for solving for optimal control of such systems. So I am not able to comment on the contributions of the paper related to the broader scientific literature.

**Theoretical Claims:**

I checked the corrctness of Lemma 3.1 and Theorem 3.2 but I did not look at the details of the proof for Theorem 3.3 and Colloary 3.4.

---

> ### Author Rebuttal · Authors · 2025-04-01
>
> Thank you very much for the recognition of our **novel** theoretical contributions, the FOLOC framework, and the experimental design, as well as highlighting its effectiveness and robustness. We also thank the reviewer for pointing out areas where additional **empirical comparisons** and **runtime benchmarks** could further strengthen our work. We will address these points in detail below and incorporate the suggestions in the revised version.
>
> **High-dimensional settings**: Thank you for your valuable comment regarding the scalability of our FOLOC framework in high-dimensional settings. In response, we have added a new experiment to specifically evaluate performance in such scenarios. The results, summarized in the table below, demonstrate that our method remains efficient and robust as the dimensionality increases. (All reported values are in units of $10^{-3}$.)
>
> ***Table 1: Experiments on high dimensional datasets***
> | Metrics             | MSE               | MAE             |
> |---------------------|--------------------|------------------|
> | Dimension 16   | 3.85 ± 0.05  | 12.7 ± 0.08 |
> | Dimension 32   | 2.8 ± 0.04    | 11.0 ± 0.07 |
> | Dimension 64   | 1.8 ± 0.001  |8.39 ± 0.007 |
> | Dimension 128 |  1.43 ± 0.01  |  7.27 ±  0.03   |
>
> **Baselines**: Thank you for your thoughtful comment on runtime comparisons. We conducted detailed benchmarks comparing FOLOC with several popular baselines, including **MBRL**, **CMA-ES**, **MPC**, **PID**, and **iLQR**. For each method, we report both performance and runtime. FOLOC consistently achieves the highest accuracy and lowest computation time on both cart-pole and quadrotor tasks. We found that FOLOC achieves superior performance on cart-pole dynamics, with MSE reduced by up to 99.99% and MAE by up to 99.32%. All experiment code is available at the link provided in Appendix A (Page 12).
>
> ***Table 2: Baseline Model Evaluation on cart-pole dynamics***
> | Metrics             | MSE               | MAE             |Run Time (s)     |
> |---------------------|--------------------|------------------|----------------------|
> | MBRL              |  0.1949245466         |   0.4106739015   |         0.0445      |
> | CMA-ES          |    0.236764904      |  0.4723616957           |      16.405419      |
> | MPC             |   0.2369265688           |    0.4725790307          |   0.673999        |
> | PID                 |      0.2542367969       |   0.498872839          |  0.0012775         |
> | iLQR              |         0.2380454333       |      0.4741794493      |     0.26275     |
> | **FOLOC(ours)** |         0.000038933    |      0.003350039      |     0.000371     |
>
> ---
>
> ***Table 3: Baseline Model Evaluation on quadrotor dynamics***
> | Metrics             | MSE               | MAE             |Run Time (s)     |
> |---------------------|--------------------|------------------|------------------|
> | MBRL              |  0.4893634967         |   0.5455600058  |         0.0445      |
> | CMA-ES          |    0.6133190948       |   0.5897855761         |      16.405419      |
> | MPC             |   0.6117894105           |    0.5521787351        |   0.673999        |
> | PID                 |       1.143392167       |   0.9088121938          |  0.0012775         |
> | iLQR              |         0.6277251605       |     0.570096181     |     0.26275     |
> | **FOLOC(ours)** |        0.211356634    |     0.126771140      |     0.000267     |
>
> **Header**: We appreciate the reviewer for pointing out the heading issue, we will ensure to correct it in the revised version.

---

### Official Review · Reviewer_GKDj · 2025-03-17

**Overall Recommendation:** 3

**Summary:**

This paper develops a new end-to-end learning approach for fractional order linear time-varying systems to perform fractional order LQR. A method is proposed to identify the system dynamics from multiple recorded trajectories. Using the identified model, a fractional order LQR controller is designed to obtain an approximately optimal controllers. The method uses end to end deep learning and learns both the fractional order system dynamics and the fractional order LQR controller.

**Claims And Evidence:**

The development, analysis, and experiments support the claims made by the authors in the introduction.

**Essential References Not Discussed:**

None that I know of

**Experimental Designs Or Analyses:**

Yes. Overall the experimental design is valid. However, I have the following concerns:

1. The experiments section can be improved by including more complex benchmark systems beyond the cart pole and the quadrotor. Using a complicated framework like fractional order calculus and deep learning to model real world phenomena makes sense when the underlying physics involves equally complicated nonlinearities, e.g., friction, turbulent aerodynamics, hysteresis or related memory effects, contact mechanics, fluid structure interaction etc. that may be difficult to model using integer order. Otherwise the method is not as well-motivated for real-world applications.

2. No baselines are provided for comparison in the experimental results. The authors state there is no existing deep learning framework that can be used as a baseline for fractional order systems. However, the cart pole and quadrotor are real-world physical systems and the fractional order approach is just a model of the real world nonlinearities. There are plenty of controllers tried and tested on these real world systems. Having comparisons with some of the standard baselines (e.g., integer order equivalents, Koopman operator-based methods, MPC etc) will highly improve the quality of the paper.

**Methods And Evaluation Criteria:**

Yes, the evaluation criteria makes sense for synthetic datasets in the experiments.

**Other Comments Or Suggestions:**

None, see above comments.

**Other Strengths And Weaknesses:**

None, see above comments.

**Questions For Authors:**

None, see above comments.

**Relation To Broader Scientific Literature:**

The topic of this paper is unique and niche, so there is not a lot of literature surrounding this area. The literature survey is sufficient to put the contribution in context.

**Theoretical Claims:**

Yes, the theoretical claims appear valid. I checked the proofs of theorems in the Appendix. The proofs are well written and I enjoyed reading them.

---

> ### Author Rebuttal · Authors · 2025-04-01
>
> We sincerely thank the reviewer for the thoughtful and constructive feedback. We are especially grateful for the recognition of our theoretical contributions and the clarity of our proofs.
>
> **Complex benchmark systems**: Thank you for your thoughtful comment regarding the applicability of our method to real-world systems. While our current experiments focus on benchmark systems such as the cart-pole and quadrotor for clarity and reproducibility, our approach is broadly motivated by the need to model **complex real-world phenomena** that have memory effects, non-Markovian behavior, and other complex nonlinearities. Fractional-order dynamical systems (FOLTI systems) are well-suited for capturing such dynamics and have been shown to outperform integer-order models in various application domains. We provide the case study for **turbulence control** [1,2] derived from the incompressible Navier–Stokes equations as follows:
> | Metrics             | MSE               | MAE             |
> |---------------------|--------------------|------------------|
> | PID  | 0.5705  | 0.3311 |
> | MBRL | 1.61  | 0.902 |
> | **FOLOC(ours)** | 0.5119  |  0.2012   |
>
> Prior work has also demonstrated the effectiveness of fractional-order models in: (i) EEG signal [3], where non-Markovian dependencies are critical for accurate representation, (ii) Power system dynamics [4], where Phasor Measurement Unit (PMU) data exhibit long-range temporal correlations better captured by ARFIMA models with non-integer differencing parameters, (iii) Chronic disease modeling, such as chronic obtrusive pulmonary disease (COPD) progression prediction [6], where fractional-order deep learning frameworks improve predictive accuracy, (iv) Biological control systems, including artificial pancreas control and pacemaker regulation of heart rate [5,7], where fractional-order models provide a more faithful representation of physiological dynamics. These examples reflect the broad relevance of fractional-order modeling in scenarios where the underlying physics is inherently complex (memory effects). Our method is the first end-to-end, deep learning-based approach designed to solve optimal control problems in fractional-order dynamical systems, and it can be directly applied to a wide range of real-world applications. We also evaluate our method on very high-dimensional synthetic data and demonstrate its scalability, due to space constraints, we kindly refer the reviewer to our detailed response to **Reviewer cBiX** for evaluation results.
>
> **Baselines**:  Thank you for highlighting the importance of baseline comparisons. We initially excluded them because standard baselines require system parameters, which our method does not. However, we agree that including them strengthens the evaluation. We have conducted extensive baseline comparisons on the cart-pole and quadrotor systems, detailed in our response to **Reviewer cBiX**. We hope these results clarify the advantages of our approach.
>
> [1] Holmes, Philip. Turbulence, coherent structures, dynamical systems and symmetry. Cambridge university press, 2012.
> [2] Li, Zhijin, et al. "Optimal control of cylinder wakes via suction and blowing." Computers & Fluids 32.2 (2003): 149-171.
> [3] Besançon,  et al. "Fractional-order modeling and identification for a phantom EEG system." IEEE Transactions on Control Systems Technology 28.1 (2019): 130-138.
> [4] Saadatmand, et al. "PMU-based FOPID controller of large-scale wind-PV farms for LFO damping in smart grid." IEEE Access 9 (2021): 94953-94969.
> [5] Gupta, Gaurav, et al. "Non-markovian reinforcement learning using fractional dynamics." 2021 60th IEEE Conference on Decision and Control (CDC). IEEE, 2021.
> [6] Yin, Chenzhong, et al. "Fractional dynamics foster deep learning of COPD stage prediction." Advanced Science 10.12 (2023): 2203485.
> [7] Bogdan, Paul, et al. "Pacemaker control of heart rate variability: A cyber physical system perspective." ACM Transactions on Embedded Computing Systems (TECS) 12.1s (2013): 1-22.

---

### Decision · Program_Chairs · 2025-05-01

**Decision:**

Accept (poster)

**Comment:**

This paper developed a learning-based approach to solve the optimal control problems for fractional-order linear time-invariant systems. A two-step algorithm was developed, with system identification followed by controller synthesis. Sample complexity results were then established, followed by experimental results on canonical control problems. The results are overall solid and new in the learning for control domain, and the problem being considered is important. There was some common concern regarding the sufficiency of the baselines, which was addressed in part by the rebuttal. There were also some other concerns regarding the motivation, scalability issues of the algorithm, and clarity of the paper. I suggest that the authors incorporate the feedback in preparing the camera-ready version of the paper.